



# Sea ice in the Baltic Sea during 1993/94–2020/21 ice seasons from satellite observations and model reanalysis

Shakti Singh[1], Ilja Maljutenko[1], and Rivo Uiboupin[1]

[1]Department of Marine Systems, Tallinn University of Technology, Estonia

**Correspondence:** Shakti Singh (ssingh@taltech.ee)

**Abstract.** This study investigates the sea ice characteristics of the Baltic Sea using Copernicus satellite and model reanalysis data products from 1993 onwards. Our primary focus is on assessing the performance of the latest Copernicus model reanalysis product in estimating ice season evolution compared to the satellite dataset. Firstly, the model reanalysis dataset is bias-corrected for further analysis. While the model estimates an earlier start to the ice season, it generally matches satellite data regarding the season's end. Additionally, we find that the model tends to overestimate ice thickness compared to ice chart-based data. Across the Baltic Sea, declining trends for the sea ice are observed. The sea ice characteristics during the recent period (2007–2021) show decreased sea ice fraction and thickness. The decrease in the sea ice thickness is over 50 % in some areas during the melting phase. Trend analysis in the study reveals a uniform pattern towards shorter ice seasons (most prominent being in Bothnian Bay with a range of approximately 1–3 days/year of decline in ice season), reduced sea ice extent (SIE) and reduced mean ice thickness (reaching up to -0.4 cm/year).

## 1 Introduction

The Baltic Sea is a semi–enclosed sea located in Northern Europe, it is recognized as a shelf sea and a marginal sea of the Atlantic. It stands as one of the most extensive brackish water bodies globally (Voipio, 1981). The sea undergoes seasonal fluctuations in ice coverage (Leppäranta and Myrberg, 2009). It stands as a crucial economic zone, hosting one of the world's densest shipping networks and significant economic interests (International Maritime Organization (IMO)). The operations in this region are greatly influenced by the presence of sea ice. Hence, the precise modeling and forecasting of the sea ice season in the Baltic Sea hold significant importance for ensuring secure navigation, particularly for maritime transport, fishing fleets, and various commercial ventures like construction of offshore wind farms. Accurate spatio–temporal observations and predictions also offer important insights into climate patterns, fluctuations in ice coverage, and the broader shifts in the region's environment. During severe winters, Finland and Estonia stand out as the only countries globally where all harbors undergo complete freezing (Jevrejeva and Leppäranta, 2002), an important occurrence that amplifies the priority placed on sea ice studies in the region. The exploration of sea ice dynamics and climatology has been a focal point in numerous studies (Leppäranta, 1981; Haapala and Leppäranta, 1996, 1997; Granskog et al., 2006; Siitam et al. 2017, Raudsepp at al., 2020). Haapala and Leppäranta in 1996 studied Baltic Sea ice season climatology, using three particular winters (normal, severe. and



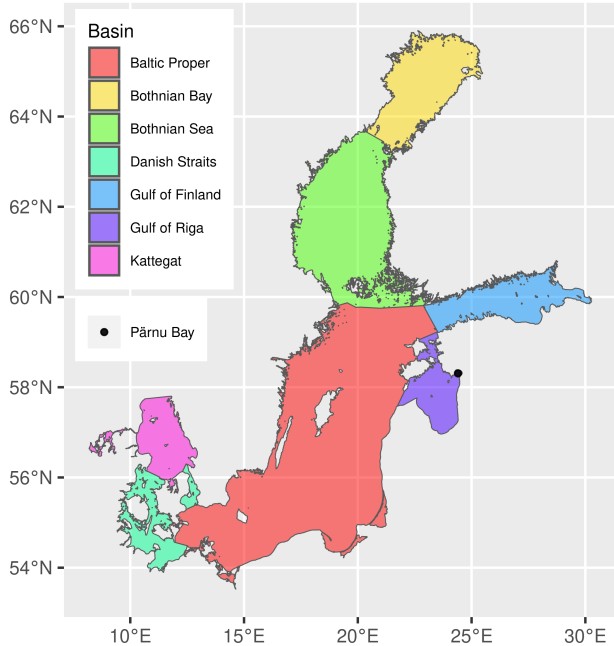

**Figure 1.** Spatial Distribution of Baltic Sea sub–basins. Figure illustrates the distinct geographical boundaries of Baltic Sea sub–basins–Bothnian Bay (BB), Bothnian Sea (BS), Gulf of Finland (GoF), Gulf of Riga (GoR), Baltic Proper (BP), Danish Straits (DS) and Kattegat–represented in unique colors for differentiation. The sub–basins are based on the PLC–6 project, and are obtained from the Helsinki Commission (HELCOM, 2018)

mild winters), and concluded that a moderate resolution ice–ocean model could reproduce the main characteristics of the ice season.

The typical duration of the ice season spans up to seven months (Vihma and Haapala, 2009), peaking in late February and early March (BACC II Author Team, 2015), when on average the ice–covered area is 45% of the total area of the Baltic Sea (Leppäranta & Myrberg, 2009). While March usually marks the onset of melting, observations in the northernmost Bothnian

Bay have recorded ice persisting until June (Leppäranta and Myrberg, 2009). So the duration of an ice season varies from some 20 to 30 days in the northern Baltic Proper to more than 6 months in the northern Bothnian Bay (Jevrejeva et al., 2004). In the Gulf of Riga sub–basin, the length of ice season observed from remote sensing data is in the range of 3–4.5 months (Siitam et al. 2017). Due to thicker ice and colder climate, ice persists longer in the Bothnian Bay compared to other sub–basins in the Baltic Sea (Pemberton et al., 2017). However, the length of the maximum ice extent period and the temporal dynamics of

ice formation or melting, the sub–basins Bothnian Bay, Bothnian Sea, Gulf of Finland, and the Gulf of Riga show a coherent pattern of the interannual variability (Raudsepp et al., 2020). BACC II Author Team, 2015 showed that the sea ice extent and thickness exhibit substantial interannual variability. Time series data analysis on maximum annual SIE and the duration of the ice season suggests a trend towards milder winters. In the 20th century, a decreasing trend of 14 to 44 days has been observed in



the length of the ice season (Jevrejeva et al., 2004). A shift in the ice season regime in 2006–2007 was reported which resulted
in ice cover and season length decrease (Pärn et al. , 2022). In the previous decade (2014–2023), a decrease in the duration of
the sea–ice melting was observed and in all sub–basins of the Baltic Sea, a rapid warming during spring was detected (Pärn
et al., 2022). Remote sensing techniques have evolved to enhance the efficacy of ice information services (Karvonen, 2004,
2013; Karvonen et al., 2005; Leppäranta and Lewis, 2007; Mäkynen and Hallikainen, 2005). The studies regarding sea ice in
the Baltic Sea have utilized different datasets. In the 2004 study, Jevrejeva et al. used the station data, whereas in a recent study
by Pärn et al., 2022, datasets from Copernicus Marine Environment Monitoring Service (Von Schuckmann et al., 2018) and
ERA5 (Hersbach, 2020) of European Centre for Medium–range Weather forecast (ECMWF) were used. Nevertheless, remote
sensing techniques offer restricted insights into sea ice thickness. Hence model reanalysis data have been utilized for the sea
ice thickness analysis.

The Copernicus Marine Service (or Copernicus Marine Environment Monitoring Service) is the marine component of the
Copernicus Programme of the European Union. It provides free, regular, and systematic authoritative information on the state
of the Blue (physical), White (sea ice), and Green (biogeochemical) ocean, on a global and regional scale. The analysis has
utilized the latest model reanalysis Baltic physics product and satellite based products (all described in the upcoming section)
from the service for the Baltic Sea. With the recent updates in the regional reanalysis of the Baltic Sea, the new model product
has improved spatial resolution and updated sea ice model (Kärna et al., 2021), which provides an improved description of the
ice state over previous model products (QUID_REAN).

**Objectives:** An integral objective of this study was to examine the spatial and temporal disparities between sea ice character-
istics from the new release of Copernicus Marine Service Baltic Sea Physics reanalysis product (BALTICSEA_MULTIYEAR
_PHY_003_011) to the satellite and ice chart–based datasets. This comparison aims to assess the accuracy of the model–simulated
data in reproducing the observed characteristics of the Baltic Sea ice. The specific objectives are (a) finding the sea ice fraction
threshold (TH_SIF) to bias correct (minimize bias) the model reanalysis dataset for Sea Ice extent (SIE) and comparing ice
thickness statistics between the model and SAR & ice charts–based datasets; (b) comparing Baltic Sea Physics reanalysis prod-
uct's sea ice season evolution characteristics with satellite dataset; (c) analyzing the characteristics and changes of ice extent
and thickness in during 1993/94–2020/21; (d) providing trend analysis of the sea ice season parameters and sea ice thickness.

## 2 Datasets

The study utilized the Copernicus Marine Sea Surface Temperature (SST) dataset from the reprocessed L4 product which
contains the Sea Ice Fraction (SIF). The Copernicus Marine SST reprocessed L4 product, known as SST_BAL_SST_L4_REP
_OBSERVATIONS_010_016 (Høyer et al., 2016) has been developed at the Danish Meteorological Institute (DMI) and offers
comprehensive, gap–free maps of sea surface temperature (SST) for the Baltic Sea region. The L4 product is made at a high
resolution of 2.4×1.44 km (0.02 °×0.02 °) using satellite data from ESA's SST CCI and Copernicus C3S projects. incorporating
an array of different sensors on satellites like NOAA AVHRRs Metop, ATSR1, ATSR2, AATSR, and SLSTR. It also combines



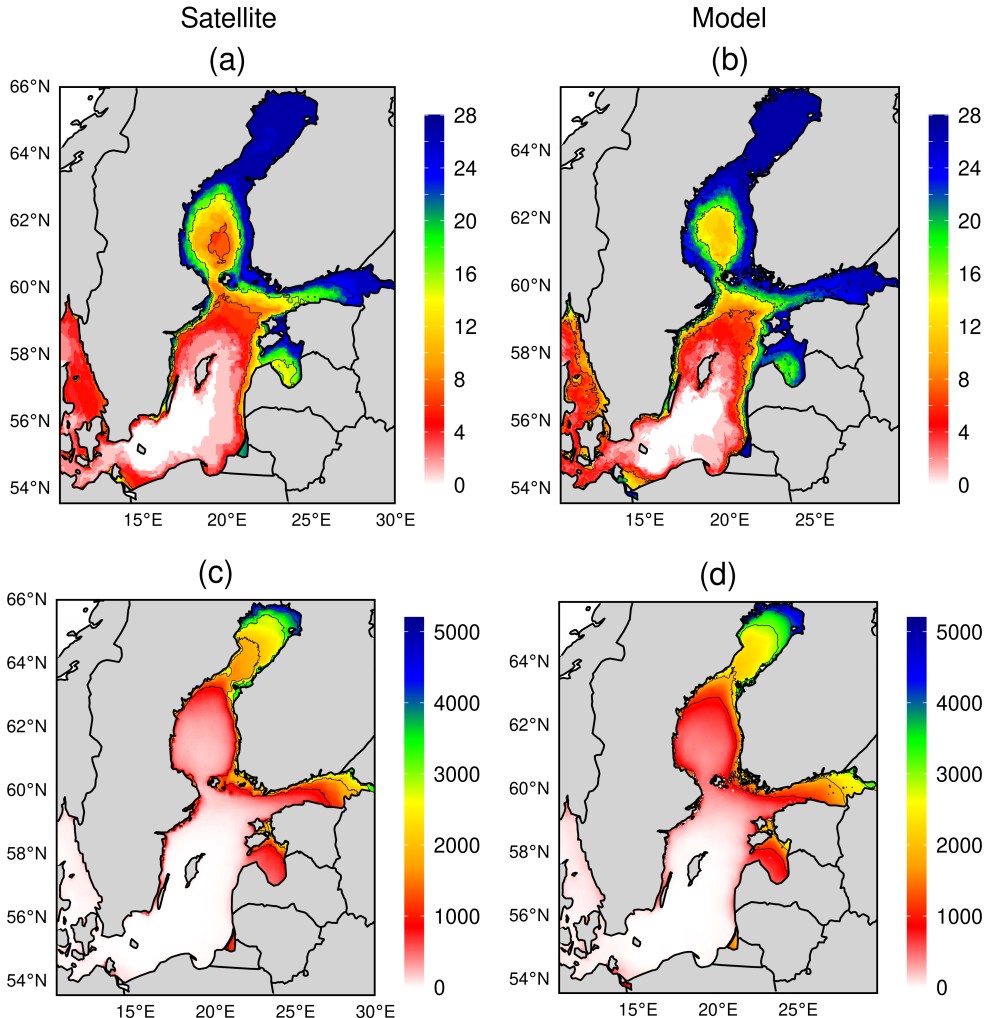

**Figure 2.** The sample size of the datasets used in the calculations, Panels (a) and (b) illustrate the count of years factored into the calculations (solid thin black contour lines are at value of 8, 16, and 24 years), while panels (c) and (d) illustrate the aggregated days considered at individual grid point (solid thin black contour lines are at value of 1000, 2000, 3000, and 4000 days). Left side panels (a & c) are for satellite data and right side (b & d) are for model data

this data with in situ measurements from the HadIOD dataset and high–resolution sea ice information from SMHI and FMI. The derived L4 sea surface temperature (SST) used in the product has been compared with different sources of in situ observations by Englyst et al. (2023). Hereafter in the study, this dataset is referred to as the satellite dataset.

The latest reprocessing product (released in March 2023), Copernicus Marine Baltic Sea Physics Reanalysis (BALTIC-

SEA_MULTIYEAR_PHY_003_011) have been used in the study. The study utilizes Sea Ice Fraction (SIF) and Sea Ice Thickness (SIT) parameters from this new dataset. The product is based on simulations using the 3D ocean–ice model NEMO



(Nucleus for European Modelling of the Ocean) regional Nemo–Nordic 2.0 configuration (Kärna et al, 2021). It offers a comprehensive reanalysis of physical conditions in the Baltic Sea available from January 1993, providing a $2\times2$ km ($0.0277$ $°\times0.0166$ °) grid resolution. The physical system assimilates satellite sea surface temperature and in situ temperature and salinity profiles. The system is forced by ECMWF ERA5 meteorology (ref). The fully coupled sea ice model SI3 addresses various aspects of sea ice dynamics, including thermodynamics, advection, rheology, and the processes of ridging and rafting. The land fast ice parametrization utilized in this model follows the approach outlined by Lemieux et al. (2016). The model comprises five ice categories and one snow category, with specified thickness bounds set at 0.45, 1.13, 2.14, and 3.67 m. In this work, the standard settings within the sea ice model, without ice assimilation, originally developed for the global ocean for defining the ice thickness categories, are used. Subsequently in the study, this dataset is referred to as the model dataset.

The Copernicus Marine Baltic Sea–Sea Ice Concentration and Thickness Charts product dataset with product id SEAICE _BAL_SEAICE_L4_NRT_OBSERVATIONS_011_004 (Karvonen et al., 2007) has been used in the study to compare the ice thickness statistics with the model dataset. This dataset has $1\times1$ km spatial resolution and is available from 1 Jan 2018 to 8 Feb 2024. The ice parameters in the product are based on SAR image and ice charts produced on a daily basis during the Baltic Sea ice season provided by FMI and SMHI. This is referred to as the SAR & ice charts–based product in the study.

The satellite dataset spans from 1993/94 to 2020/21, the 2012/13 season was excluded (due to the absence of satellite data for the entire season), resulting in a study period covering 27 ice seasons. However, due to the absence of sea ice (absence here refers to sea ice below TH_SIF, explained in the subsequent section) at specific grid points in the Baltic Sea during different years, the data sample size varies across locations. Figure 2 shows this variation, where (a) and (b) indicate the sample count of years used, while (c) and (d) represent the sample count of days used in the study. So the calculations of the ice season parameters were based on the data sample size shown in Fig. 2. The procedure employed in the study for calculation of these parameters is given in the subsequent methodology section.

## 3 Methodology

The analysis of sea ice statistics in the Baltic Sea region relied on satellite and model datasets, which were used for the calculation of the annual ice season parameters for the study period. For the satellite dataset, a sea ice fraction threshold (TH_SIF) of 0.15 was applied, while the model dataset utilized a TH_SIF of 0.20. The reasoning for choosing these thresholds is given in the subsequent section.

Identification of the onset date of sea ice formation at each grid point ($i$, $j$) was established when the daily mean SIF ($i$, $j$, $d$, $y$) equaled or surpassed the designated TH_SIF from October 1st to March 31st. The **first day (FD)** is defined for each year y as the lowest Julian day index (*JDI*) d meeting the TH_SIF criteria within this time range [1]. Conversely, the conclusion date of the sea ice period occurred at grid points where the TH_SIF criteria were last met from January 1st to September 30th. The **last day (LD)** is calculated as the highest value of d meeting the criteria within this time range [2]. At each grid, the FD and LD of sea ice are only calculated when there was sea ice observed at those grids (here the occurrence of sea ice for each grid is defined using TH_SIF), hence each grid has a different number of sample sizes (Fig. 2a, 2b) for the calculations of the



FD and LD of sea ice. The **total days (TD)** of sea ice is the duration between the FD and LD of sea ice occurrence, broadly representing the **sea ice season** [3]. Conversely, the **number of days (ND)** of sea ice specifically accounts for instances within this season that meet the TH_SIF criteria [4].

$$\text{FD}(i,j) = \frac{1}{n} \sum_{y=1}^{n} \min(JDI(i,j,y)) \tag{1}$$

Where $JDI = \{d|\ d \text{ in Range(October 1st, March 31st)}, \text{SIF}(i,j,d,y) \geq \text{TH\_SIF in year } y\}$

$$\text{LD}(i,j) = \frac{1}{n} \sum_{y=1}^{n} \max\big(JDI'(i,j,y+1)\big) \tag{2}$$

Where $JDI' = \{d|\ d \text{ in Range(January 1st, September 30th)}, \text{SIF}(i,j,d,y) \geq \text{TH\_SIF in next year } y+1\}$

$$\text{TD}(i,j) = \frac{1}{n} \sum_{y=1}^{n} \big(\max\big(JDI'(i,j,y+1)\big) - \min\big(JDI(i,j,y)\big) + 1\big) \tag{3}$$

$$\text{ND}(i,j) = \frac{1}{n} \sum_{y=1}^{n} \text{count}(JDI''(i,j,y)) \tag{4}$$

Where $JDI'' = \{d|\ d \text{ in Range(October 1st, September 30th)}, \text{SIF}(i,j,d,y) \geq \text{TH\_SIF in year } y\}$

Comparison between two distinct periods, 1993/1994–2006/2007 and 2007/2008–2020/2021 (hereafter referred to as the preceding period, and recent period respectively), was conducted using 2D histograms. This approach aimed to study the changes in sea ice statistics over the recent 14 year period compared to the preceding 14 year span. Analytical methods included the application of linear regression techniques to discern trends between sea ice parameters within the dataset (the lm function from R). Additionally, the determination of the maximum extent involved computing the integral over the area of the

grids where the ice fraction exceeded the designated TH_SIF, serving as an indicator of peak ice coverage during the analyzed periods. A linear trend analysis of the sea ice season parameters and ice thickness has been performed across all the grid points on the Baltic Sea. The data sample size is notably higher in the model dataset than in the satellite dataset across the Baltic Sea (Fig. 2). This suggests a more frequent occurrence of sea ice fraction exceeding the threshold (TH_SIF) in the model data compared to the satellite data.

**4  Comparison of the model and the satellite dataset**

The quality information document (QUID) for the Baltic Sea reanalysis product (BALTICSEA_MULTIYEAR_PHY_003_011) validates simulated sea ice statistics against the reference product SEAICE_BAL_SEAICE_L4_NRT_OBSERVATIONS_011





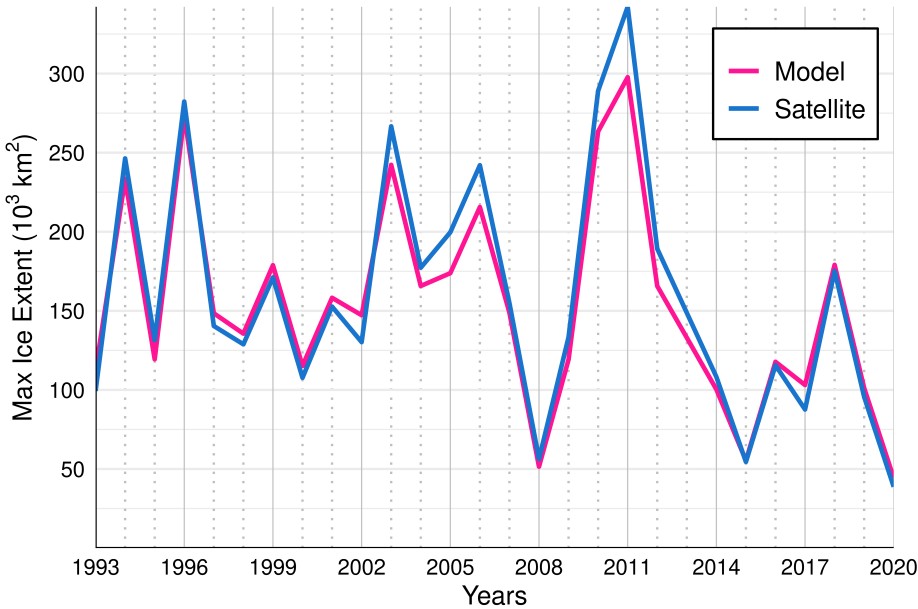

**Figure 3.** Time series of maximum sea ice cover extent (in $10^3$ km$^2$) in the Baltic Sea across years (1993 to 2020): A comparison between Model (TH_SIF 0.20) and Satellite (TH_SIF 0.15) datasets

_004 (Karvonen et al., 2007) & SMHI ice charts for sea ice concentration and thickness. In the QUID file, the ice extent (mean over all daily values) had a correlation coefficient of 0.94, and the coefficient for ice volume was 0.64. Furthermore, the

maximum ice volume estimated from the modeled ice thickness would be around two times greater than the ice volume derived from ice chart data. Concerning our study, a further relevant comparison has been done in the Sect. 4.1 & 4.2, to minimize the bias for subsequent analysis in the study.

## 4.1   Sea Ice Extent

The sea ice extent (SIE) is calculated by taking an integral over an area of each grid that satisfies the ice fraction threshold

criteria. The analysis was aimed at finding out the differences in the SIE between the model reanalysis and the satellite dataset. A threshold value of 0.15 sea ice fraction (SIF) has been used traditionally when computing the sea ice extent (Parkinson, 1987). Hence, in satellite observations, the key TH_SIF was set at 0.15 to calculate SIE. It serves as a crucial point of reference for the study, as the study aimed to determine the appropriate TH_SIF within the model dataset that corresponded accurately to the 0.15 TH_SIF utilized in satellite observations. The comparative analysis of temporal evolution during the ice season for

both of these datasets is done for that purpose. The set of SIE estimations was calculated from the model dataset using different SIF_TH and compared to the satellite dataset by means of robust statistics (Table 1). The correlation coefficient values for all the estimations are very high, hence, we focus on the combination that shows the lowest or close to the lowest Bias and RMSE values, which happens to be the combination with a sea ice fraction TH_SIF value of 0.20 for the model dataset and 0.15 for





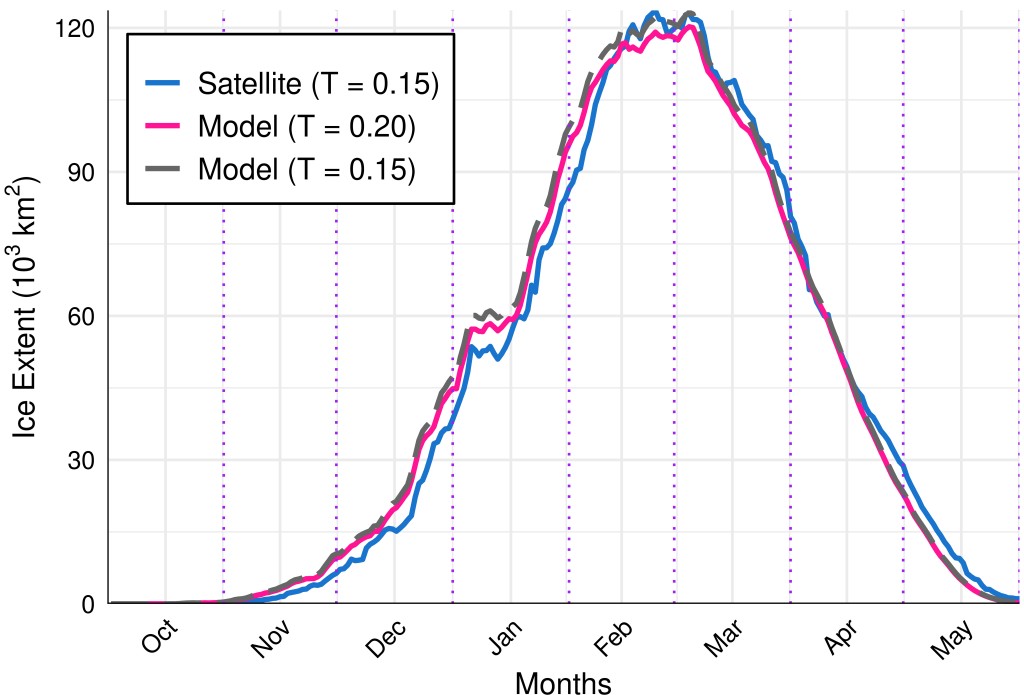

**Figure 4.** Sea ice season evolution of daily mean sea ice extent (in $10^3$ km$^2$) in the Baltic Sea (1993/94 to 2020/21): A comparison of Model (TH_SIF 0.20) and Satellite (TH_SIF 0.15) datasets

**Table 1.** Comparison of sea ice extent calculated using different threshold values of SIF for the model dataset is shown by means of statistical methods such as correlation coefficients (CC), root mean square error (RMSE), and mean bias.

| TH_SIF (Satellite) | TH_SIF (Model) | CC | RMSE (km$^2$) | Bias (km$^2$) |
|---|---|---|---|---|
| 0.15 | 0.10 | 0.993 | 6574 | 3476 |
| 0.15 | 0.15 | 0.994 | 4887 | 1783 |
| 0.15 | 0.20 | 0.995 | 3991 | 416 |
| 0.15 | 0.25 | 0.996 | 3810 | 764 |
| 0.15 | 0.30 | 0.997 | 4193 | -1820 |

satellite observations. This combination of TH_SIF for the datasets shows the minimum bias of 416 km$^2$ and the second–lowest
RMSE value at 3991 km$^2$, in close affinity to the lowest RMSE value (Table 1).

TH_SIF of 0.15 for the model dataset, provides more accurate estimates of maximum SIE (Fig. 4), while TH_SIF of 0.20
is more suitable for temporal characteristics of the ice season, which is the primary focus of the study, and hence TH_SIF of
0.20 is used for further analysis in the subsequent sections. Using TH_SIF of 0.20 may not serve as the optimal approximation



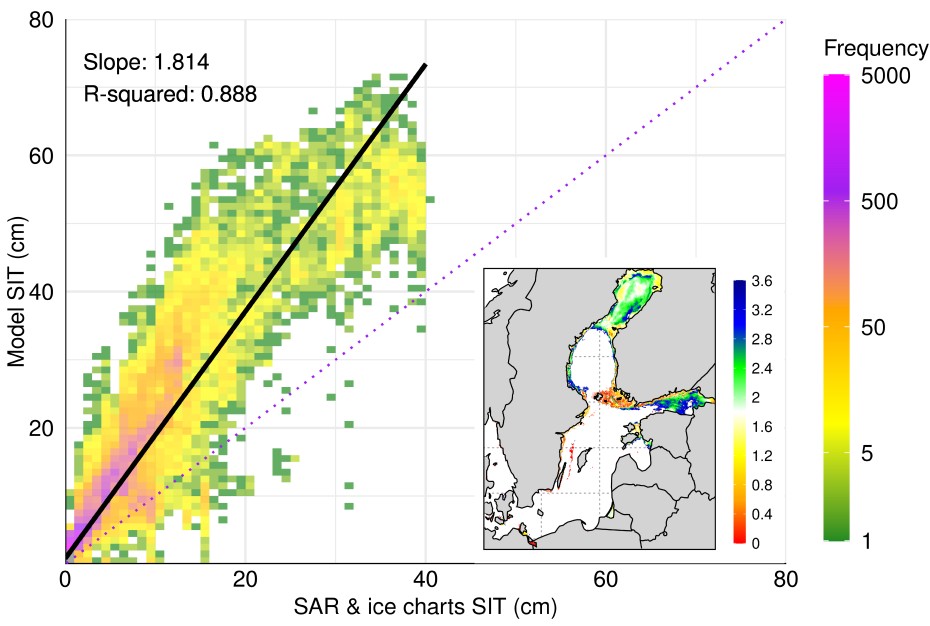

**Figure 5.** Time averaged sea ice thickness (in cm): SAR & ice charts vs model based dataset. The subplot shows the ratio of time averaged sea ice thickness from model to the SAR & ice charts value

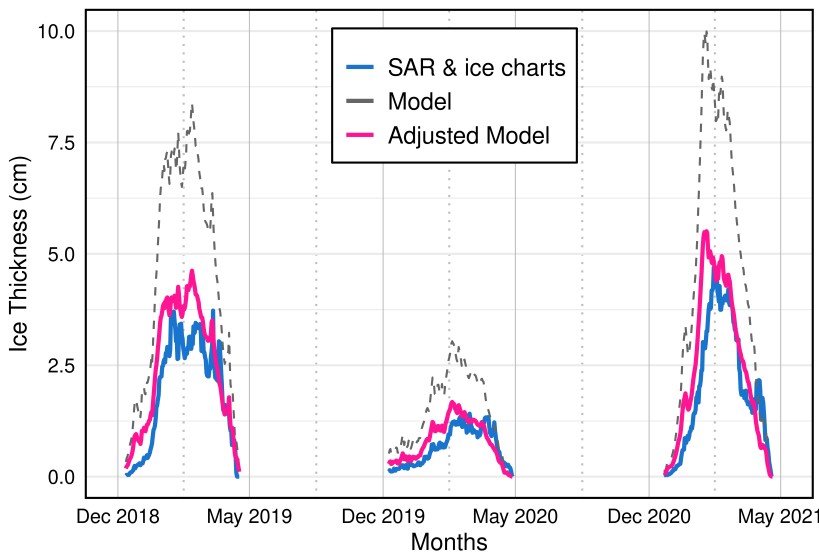

**Figure 6.** Time series of sea ice thickness spatially averaged over the Baltic Sea for the model and SAR & ice charts based datasets




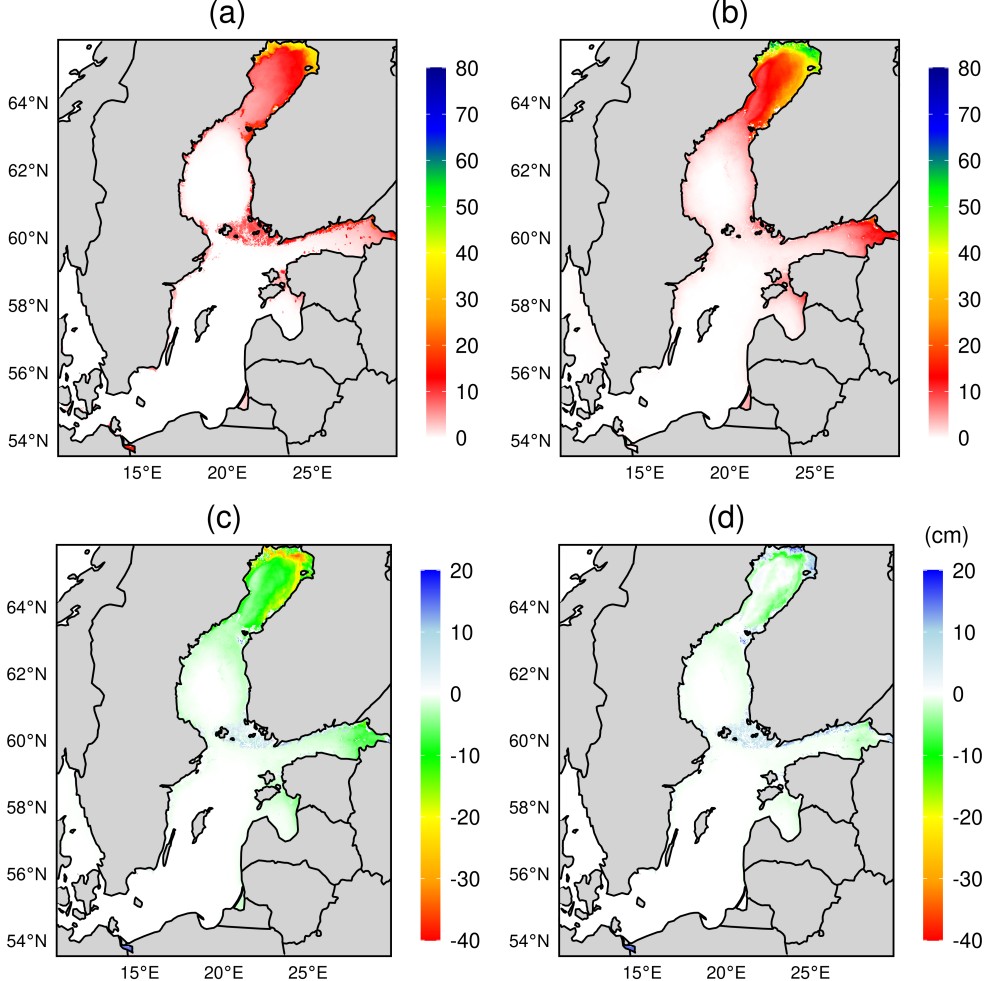

**Figure 7.** Sea ice thickness, averaged over the three ice seasons (a) from SAR & Ice charts (b) from model (c) model subtracted from SAR & Ice charts (d) adjusted model subtracted from SAR & Ice charts

for the maximum extent, but we sought to assess its effectiveness in approximating the maximum extent on an interannual

scale. This examination involved plotting the annual maximum extent for each year within the study period (Fig. 3). Despite its shortcomings, this specific combination still exhibits relative consistency in approximating the maximum extent values across the inter–annual scale. We have used some sea ice season parameters in the subsequent Sect. 5.1 (these parameters have already been defined in the methodology section) to further strengthen our understanding between the two datasets and find out the disparities after using the more suitable TH_SIF for model reanalysis dataset to better depict the satellite observed sea ice

season characteristics in the Baltic Sea.





## 4.2 Sea Ice Thickness

The ice thickness data statistics from the model have been compared to SAR image & ice chart based product for three available ice seasons (2018/19, 2019/20 and 2020/21). The model versus observations comparison (Fig. 5) shows that the model overestimates the ice thicknesses by a factor of 1.814, which is consistent with inferences drawn from the QUID file of the
model product regarding ice volume. This factor is uniform across different thickness ranges. The model tends to overestimate ice thickness more offshore than near the coasts (Fig. 5 subplot). Despite a high correlation (0.936) the model shows approximately a two–fold overestimation in average thicknesses. The corrected ice thicknesses show a closer resemblance to SAR & ice charts based thicknesses (Fig. 6). After applying the correction factor, the agreement between the model's and SAR & ice charts based ice thicknesses has improved, resulting in minimal differences across the various sub–regions (Fig. 7d).

## 5 Results

### 5.1 Sea Ice Days Characteristics : Satellite vs Model Dataset

The spatial distribution of temporal characteristics of the Baltic Sea ice climatology are shown in Figure 8 and a summary of each sub–basin in Table 2. Until the end of December, sea ice is predominantly observed in the northern BB sub-basin (Fig. 8a), and it persists up to May in the northernmost parts (Fig. 8d), resulting in an ice season that lasts approximately 5–6 months
in that region (Fig. 8g). Hence, these parts have the largest sample size of the sea ice data for the analysis (Fig. 2a, 2c). The sea ice was observed during nearly all the years of the study period for the BB and the eastern GoF sub–basin (Fig. 2a). The Bias correction is helpful in correcting or minimizing these errors. For the sea ice, bias correction has been mostly focused on correcting the total sea ice area or extent (Fučkar et al., 2014; Krikken et al., 2016). Bias correcting the models to improve their predictive capabilities of the Baltic Sea ice becomes important for all the winter activities (mentioned in Introduction) in
the region. For example, Pärnu Bay and the region between Estonia and its two major islands, have a longer sea ice season compared to the nearby areas (Fig. 8g). Except for a few years, the sea ice has also been consistently present (more than 80 % of the years) in these parts (Fig. 2a).

The differences between model and satellite show high spatial variability, specifically in the FD estimates. The model consistently estimates that sea ice starts to form earlier in most parts of the Baltic Sea compared to satellite data (Fig. 8c).
This discrepancy is particularly pronounced in the BB, GoF, and GoR sub–basins. On the other hand, the model dataset is more accurate in estimating the end of the sea ice season (Fig. 8f). Modeled LD align better with satellite observations when compared to the FD estimates (Fig. 8f, Table 2). Therefore, earlier FD estimates are causing the overestimation in the modeled TD. The largest discrepancies in TD, similar to the FD between model and satellite, are in the BB and GoF sub–basins (Fig. 8i).

The ND of the sea ice differs the most from TD in the BS and BP sub–basins (Fig. 8g, 8j), as it depends on the sea ice dynamics within the ice season, such as sea ice drift. The ND disparities between datasets (Fig. 8l) show that the BB and GoF sub–basins exhibit the largest differences. The basin specific ice temporal characteristics (Table 2) are calculated by

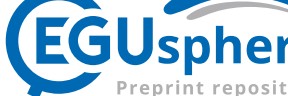



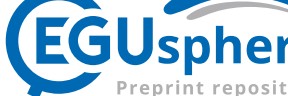

**Figure 8.** Map distribution of the ice season parameters, the FD of sea ice (a, b, c), the LD of sea ice (d, e, f), the TD of sea ice (g, h, i) and the ND of sea ice (j, k, l). The panels on the left column (a, d, g, j) are from the satellite dataset, the panels in the center column (b, e, h, k) are from the model dataset, and the panels on right column (c, f, i, l) are the differences of the model values from the satellite. The thin solid black contour lines mark the end of each month in the FD and the LD plots while these lines denote the values at 30–day intervals for the TD and ND plots



**Table 2.** Comparative Analysis of Satellite and Model Parameters in Baltic Sea sub–basins. The table compares parameters–FD, LD, TD, and ND–across distinct sub–basins in the Baltic Sea region

| Parameters | First Day | | Last Day | | Total Days | | Number of Days | |
|---|---|---|---|---|---|---|---|---|
| | Satellite | Model | Satellite | Model | Satellite | Model | Satellite | Model |
| Bothnian Bay | 31 Dec | 22 Dec | 25 Apr | 29 Apr | 117 | 131 | 103 | 115 |
| Bothnian Sea | 31 Jan | 25 Jan | 24 Mar | 24 Mar | 54 | 61 | 33 | 38 |
| Gulf of Finland | 19 Jan | 9 Jan | 1 Apr | 31 Mar | 73 | 83 | 56 | 65 |
| Gulf of Riga | 22 Jan | 13 Jan | 22 Mar | 17 Mar | 60 | 64 | 41 | 43 |
| Baltic Proper | 12 Feb | 10 Feb | 1 Mar | 2 Mar | 20 | 23 | 8 | 9 |

horizontally averaging sea ice day parameters over the regions shown in Fig. 1. The FD average has a consistent 9–10 day offset towards the early onset date in the model within the BB, GoF, and GoR sub–basins compared to satellite FD. When considering the LD across datasets, a somewhat consistent pattern of ±5 days difference is evident. Consequently, the TD exhibits the most pronounced differences in the BB and GoF sub–basins reaching 14 and 10 days respectively, meanwhile, the BS sub–basin exhibits a noticeable 7 day difference. The ND of sea ice has the largest differences of 12 and 9 days in the BB and GoF sub–basins as well. Therefore, the model consistently overestimates the ice season, mostly due to its tendency to estimate an earlier onset of the ice season.

Contrary to other sub–basins, GoR sub–basin does not display significant discrepancies in ice season length between the two datasets, owing to the model's estimates indicating a forward shift in both the FD and LD of the sea ice season. In line with trends observed in other parameters, the ND of the sea ice value exhibits the highest differences in BB and GoF sub–basins of 12 and 9 days respectively, while the other sub–basins indicate a closer resemblance between the model dataset and satellite observations for the parameter.

## 5.2 Sea Ice Fraction and Sea Ice Thickness Changes: Model Dataset

Based on the regime shift detected by Pärn et al. (2022) we split our study period into two parts: 1993/94–2006/07 (preceding period) and 2007/08–2020/21 (recent period), which allows us to analyze how the sea ice characteristics had changed in these periods. The average max extent has been less in the recent period, and the length of the ice season has also been shorter compared to the preceding period in the Baltic Sea (Fig. 9). The max SIE has decreased from approx $141 \times 10^3$ km$^2$ during the preceding period to $109 \times 10^3$ km$^2$ in the recent period (Fig. 9).

The decrease in SIE is more prominent during the melting period (after the peak SIE), as compared to the ice formation period (before the peak SIE) (Fig. 9). To separately study these changes during sea ice formation and melting, we have divided the ice season into two phases: the freezing phase and the melting phase, as December–January–February (DJF) and March–April–May (MAM) respectively. The date of maximum SIE day varies from year to year, however, its date fluctuates




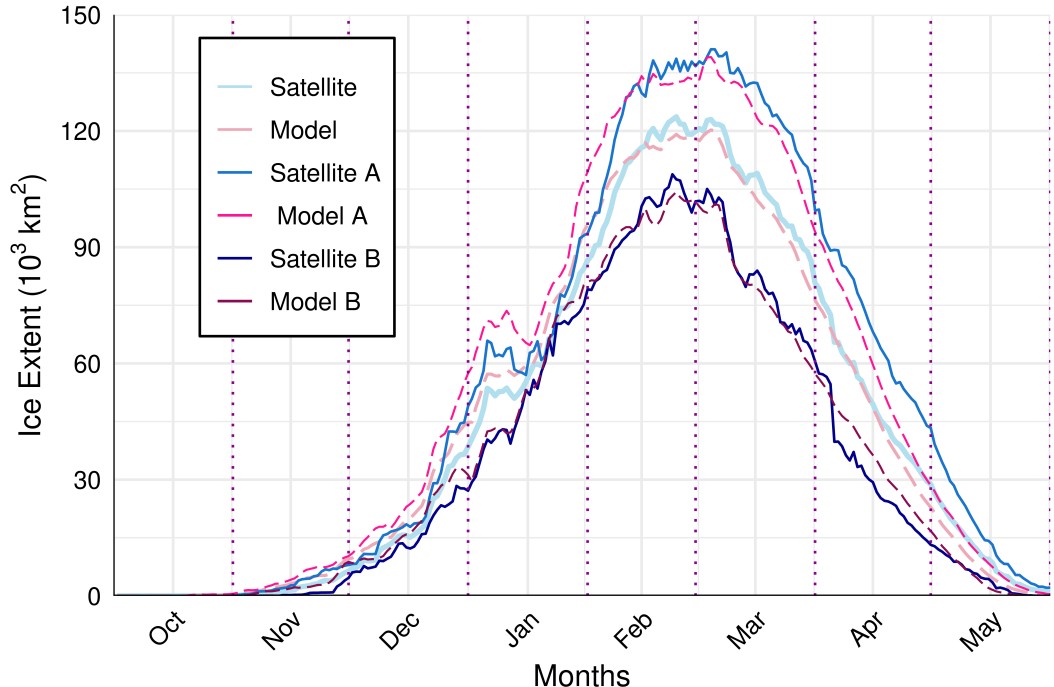

**Figure 9.** Sea ice season evolution of daily average Baltic Sea ice extent (in $10^3$ km$^2$) from 1993/94 to 2020/21: Model (dashed) versus Satellite (solid) datasets, Assessing three periods 1993/94–2006/07, 2007/08–2020/21 (referred as A and B respectively), and the complete 1993/94–2020/21 period

**Table 3.** Sub–basin averaged values of Ice Fraction and Ice thickness for both the periods (A & B) during the Freezing phase (DJF) and the Melting phase (MAM). Within the table context, A refers to the preceding period (1993/94–2006/07), while B represents the recent period (2007/08–2020/21)

| | DJF | | | | MAM | | | |
|---|---|---|---|---|---|---|---|---|
| **Sub–basin** | **SIF** | | **SIT (cm)** | | **SIF** | | **SIT (cm)** | |
| Period | A | B | A | B | A | B | A | B |
| Bothnian Bay | 0.62 | 0.50 | 17.25 | 12.18 | 0.67 | 0.52 | 34.74 | 21.54 |
| Bothnian Sea | 0.20 | 0.14 | 4.13 | 2.87 | 0.20 | 0.11 | 5.62 | 3.36 |
| Gulf of Finland | 0.44 | 0.27 | 10.14 | 5.35 | 0.37 | 0.21 | 12.13 | 5.73 |
| Gulf of Riga | 0.29 | 0.20 | 5.73 | 3.69 | 0.20 | 0.13 | 5.68 | 3.25 |
| Baltic Proper | 0.03 | 0.02 | 0.72 | 0.50 | 0.02 | 0.01 | 0.50 | 0.22 |

around the end of February to the beginning of March (Fig. 4). Ice fraction and thickness decreases are higher in the recent period during the melting phase compared to the freezing phase, specifically in the BB sub–basin (Fig. 10d, 10e).





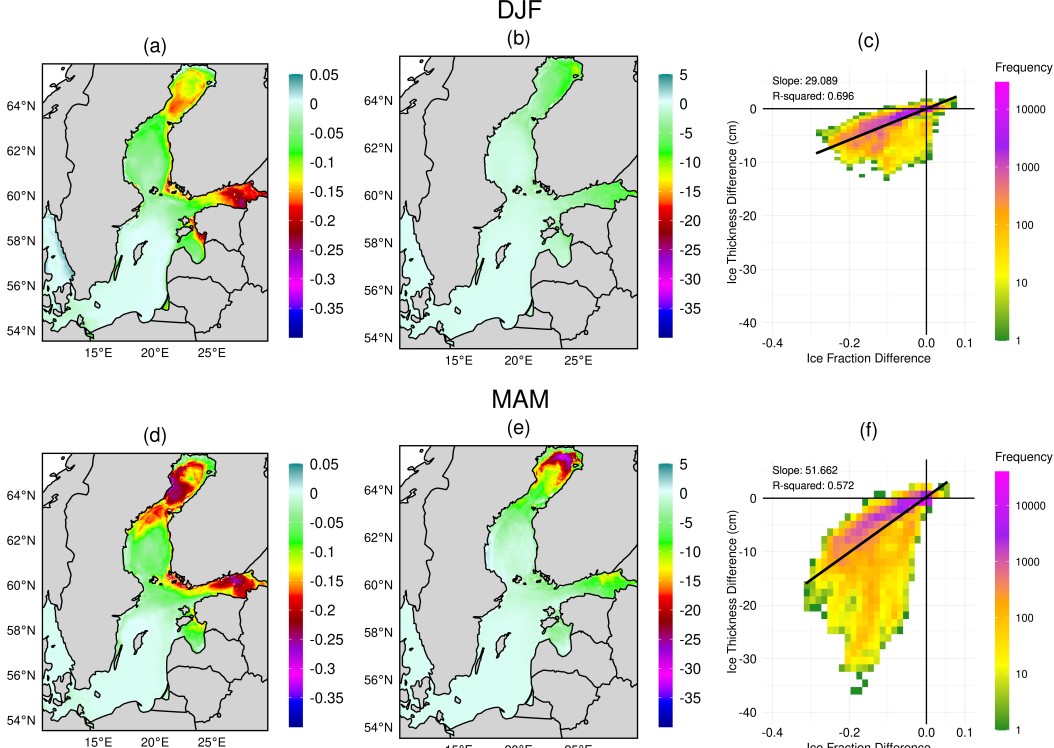

**Figure 10.** The changes of the SIF and SIT during the recent period 2007/08–2020/21 compared to preceding period 1993/94–2006/07, Panels (a) and (d) show the sea ice fraction (SIF) differences; (b) and (e) the sea ice thickness differences (SIT, in cm); while (c) and (f) shows the 2D histograms plotted for the fraction vs thickness differences, shaded according to their frequency. Linear regression lines are plotted along with their respective slope and r–squared values. Upper panels are for the winter season (DJF), and lower panels are for spring season (MAM)

**Table 4.** Changes in sub–basin averaged values of Ice Fraction and Ice thickness in the recent half compared to the preceding during the Freezing phase (DJF) and the Melting phase (MAM)

| Sub–basin | DJF | | | | MAM | | | |
|---|---|---|---|---|---|---|---|---|
| | SIF Difference | | SIT Difference (cm) | | SIF Difference | | SIT Difference (cm) | |
| Bothnian Bay | -0.12 | 19 % | -5.07 | 29 % | -0.15 | 23 % | -13.20 | 38 % |
| Bothnian Sea | -0.06 | 31 % | -1.26 | 31 % | -0.09 | 43 % | -2.26 | 40 % |
| Gulf of Finland | -0.16 | 37 % | -4.79 | 47 % | -0.16 | 44 % | -6.40 | 53 % |
| Gulf of Riga | -0.09 | 32 % | -2.04 | 36 % | -0.07 | 36 % | -2.43 | 43 % |
| Baltic Proper | -0.01 | 29 % | -0.22 | 31 % | -0.01 | 47 % | -0.28 | 56 % |





The coherence of the SIF and SIT changes has been shown for the freezing and melting periods in Figs. 10c and 10f. Although the R squared value for both regression lines (which shows how well the data points fit the regression line) is not significant, it still provides us with some surface level information on the sea ice characteristics of the Baltic Sea. The slopes

of regression lines suggest that ice thickness has generally decreased more rapidly with a decrease in ice fraction during the melting phase as compared to the freezing phase. From the approximated slopes of regression lines (cm per % of fraction), ice thickness decrease during the melting phase is approximately 0.52 cm as opposed to a decrease of 0.29 cm for freezing phase per % change of fraction (Fig. 10c, 10f). The sea ice fraction values and their decline in recent period show similar trends for both of the ice season phases (Table 3). Concerning sea ice thickness, the melting phase displays higher ice thickness

in the BB sub–basin during the preceding period and notably the most substantial decrease in the sub-basin (Fig. 10e). This occurrence could be attributed to the presence of thicker ice in the BB sub–basin during the melting phase, which could explain the pronounced reduction in ice thickness observed in that area. In percentage terms, even though the melting phase exhibits thicker ice during the preceding period (making it a larger reference value for percentage change), a higher thickness decrease of 38 % during the melting phase compared to a 29 % decrease during the freezing phase in the BB sub–basin is observed. The

melting phase has seen a larger decrease of ice fraction and ice thickness compared to the freezing (winter season) phase across all the sub–basins, with the most significant being in the GoF sub–basin, where during the spring season, the ice thickness has reduced to less than half of its value (Table 4). The BP and GoR sub–basins changes reflect the changes in a limited number of winters (Fig. 2) and consequently, particularly in the BP sub–basin, there is a significant reduction in sea ice percentage–wise, but in absolute terms, the decrease is relatively insignificant (Table 4).

## 5.3 Trend analysis of sea characteristics

A linear trend analysis has been performed on each grid cell of the Baltic Sea over the study period for the parameters FD, LD, TD, ND of sea ice and the ice thickness. The resulting spatial distributions of the linear trend across the Baltic Sea are shown in Fig. 11 and Fig. 12. The FD of sea ice in the Baltic Sea shows an increasing trend, which is statistically insignificant at 95 % significance level for the most parts of the Baltic Sea (Fig. 11a). However, in the case of the LD of sea ice, a decrease

of approximately 1–2 days per year above 95 % significance level is observed in almost all parts of BB sub–basin area and in some parts of the GoF sub–basin as well (Fig. 11b). Hence, a decrease of approx. 1–3 days per year in TD and ND of sea ice parameters is observed in roughly the same areas. The linear trends of the annual mean ice thicknesses averaged over ice season (November to May) show that the areas of the northern BB sub–basin have the highest and most significant (over 95 % significance level) decreasing trend of order up to approx. 0.4 cm/year (Fig. 12a). The BB has the highest thickness of sea

ice across all the sub–basins (Table 3). The linear trends that are normalized with their respective mean ice thickness show the ice thickness trend in terms of average percentage decrease. The normalized trend values have been uniform over the BB sub–basin standing at approximately -5 percent.





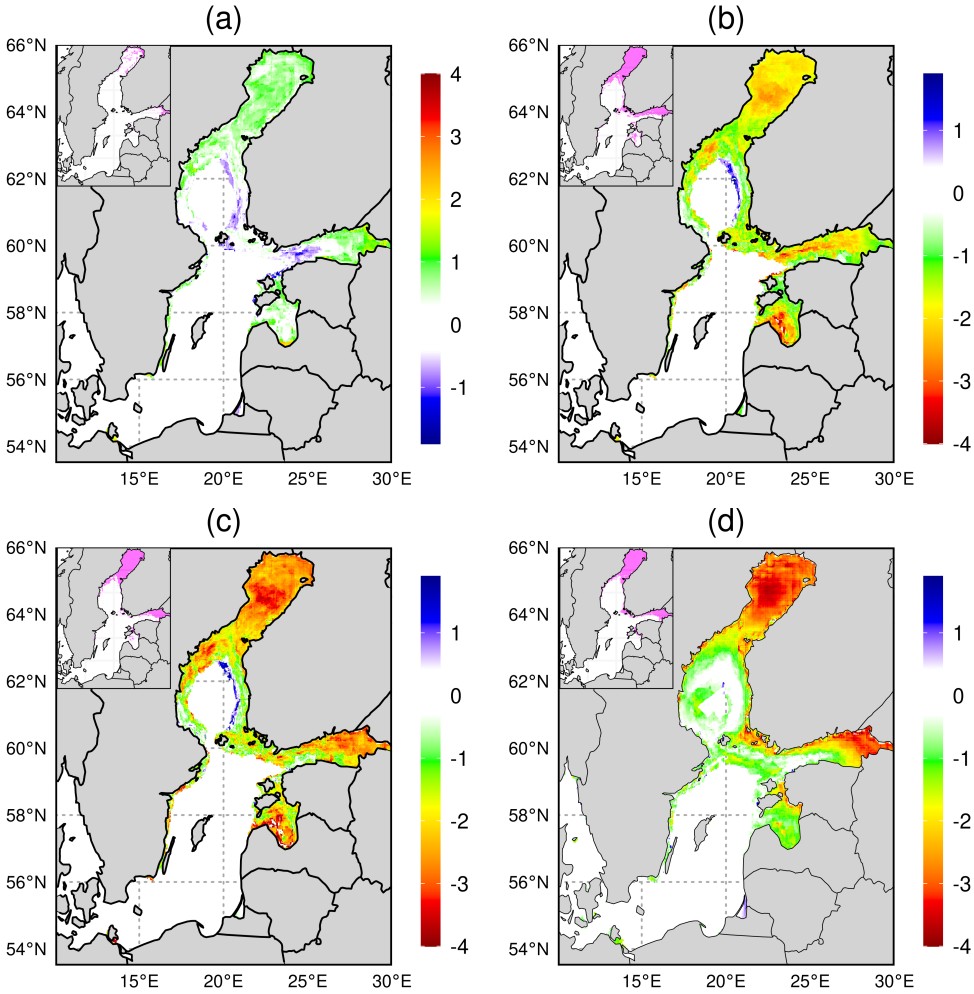

**Figure 11.** Linear trend for (a) FD (b) LD (c) TD and (d) ND of sea ice (in days/year) during 1993/94 to 2020/21 ice seasons. In the top left subplot, magenta color signifies 95 % significance level

## 6   Discussion

The state of sea ice is important for many stakeholders (Wagner et al., 2020). Monitoring the extent and thickness of Baltic
ice cover, as well as the duration of the ice season, has been crucial for navigation and travel for centuries. For example, in Finland over 80 % of international trade relies on maritime routes, and ships often require icebreaker assistance for 3 to 6 months each winter (Vihma and Haapala, 2009). Severe sea ice conditions have the potential to substantially disrupt the flow of busy Baltic maritime traffic (Löptien and Axell, 2014). The duration of ice cover in the Baltic Sea significantly impacts spring biota activity, altering the timing of spring blooms and the composition of phytoplankton species (Klais et al., 2017a;
Klais et al., 2017b; Pärn et al., 2021), thereby affecting nutrient cycles and ecosystem dynamics (Klais et al., 2013), which may



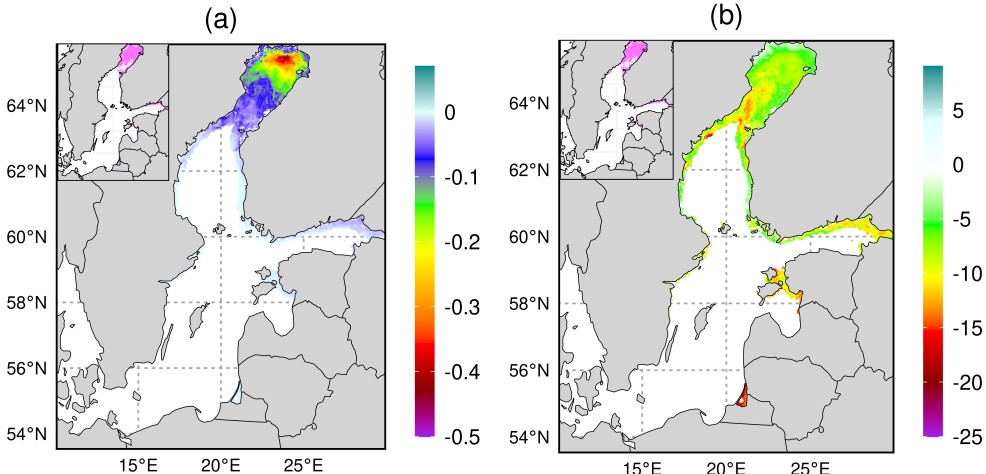

**Figure 12.** Linear trend for (a) ice thickness (in cm/year) (b) ice thickness (in %) for mean of Nov–May ice thickness from 1993/94 to 2020/21. In the top left subplot, magenta color signifies 95 % significance level

have significant implications for fisheries industries. To address these needs of various user groups the Digital Twins and their applications for European waters (incl. Baltic Sea) are being developed under the Destination Earth and EDITO (European Digital Twin of the Ocean) initiatives. Thus, accurate information and analysis of sea ice conditions/processes in the Baltic Sea provided in current study over different spatio-temporal scales (from operational monitoring to climate analysis) is also relevant

for the development of Digital Twins and related impact models (**Åström et al. 2024**). The improved sea ice information that can be ingested into the Digital Twins applications and tools enable to carry out what-if scenario analyses and thus improve the planning of offshore activities.

Bias correcting the models to improve their predictive capabilities of the Baltic Sea ice becomes important for all the winter activities (mentioned in previous paragraph) in the region. For the sea ice, bias correction has been mostly focused on

correcting the total sea ice area or extent (Fučkar et al., 2014; Krikken et al., 2016). In our study the latest Copernicus Baltic Sea Physics reanalysis product is bias corrected (using a different SIF threshold value) for the mean season evolution of the sea ice extent. For sea ice thickness, satellite product also has uncertainties, and the ice maps only take into account the visible ice thickness from ships when estimating offshore ice thickness. Moreover, they do not contain the additional volume of deformed ice (Raudsepp et al., 2019). Using the SAR imagery & ice charts based Copernicus product for ice thickness could be more

suitable, however, it is only available from 2018 onwards. Hence, the model ice thickness was bias corrected using this dataset for the analysis.

In the study by Jevrejeva et al. (2004), the long–term time series of date of freezing, breakup, number of days with ice, and maximum annual ice thickness of landfast ice in the Baltic Sea were examined using the station data, and the results provided insights into complicated variability in ice conditions. Their 100 year time series showed a general trend toward reduced ice

conditions, with the largest change being the length of the ice season, which has decreased by 14–44 days during the 21st





century. Instead of station data (which are mostly onshore), the satellite data provide coverage over the whole Baltic Sea, where we observe a reduced ice season length (on average approx. 1–3 days/year) and decreased maximum SIE (by approx. $32\times10^3$ km$^2$) in the recent period (2007/08–2020/21) of our study. As compared to the Jevrejeva et al. (2004) where most of ice season length analysis was done for coastal locations, the current study extends that knowledge to offshore areas as well, for the period 1993/94–2020/2021. The highest reduction in the ice season is also observed at the offshore areas.

The maximum ice extent of the Baltic Sea typically occurs in late February and early March (BACC II Author Team, 2015). It is also supported by our analysis of mean daily extent during ice season, as we observe the mean max SIE in the Baltic Sea during the same time. Due to the larger decrease in SIE after the peak, the changes have been studied for the freezing phase (DJF) and the melting phase (MAM) separately.

A common trend observed is that both the sea ice fraction (correspondingly sea ice extent) and sea ice thickness have decreased in the recent period during both of the ice season phases, specifically, the changes are larger during the spring season, which is in line with the study by Pärn et al. (2022), where they have observed a regime shift during spring time beginning in 2008 in all the basins of the Baltic Sea. For the BB sub–basin, the reduction in sea ice thickness during the melting phase is more pronounced in absolute terms compared to the decrease in sea ice fraction in the study. The possible causes could be partly attributed to higher sea ice thickness values in the BB during the melting phase in the preceding period of the study. There was no general conclusion concerning maximum annual ice thickness from Jevrejeva et al. (2004), while some sites (Kemi and Kihnu) showed an increase, most time series were characterized by a decreasing trend. During our study period, we observed a significant uniform decreasing trend (about 5 %) in the mean SIT on the northern BB sub–basin, reaching up to 0.4 cm/year in some parts. The value is dependent on the months selected for the annual ice thickness mean. Climatic conditions during the winter months (December–February) largely control the development of the ice cover in the Baltic Sea (Omstedt and Chen, 2001). The reduced ice conditions trends in the Baltic Sea during the 20th century were also related to a warming trend in winter air temperatures over Europe (Jevrejeva et al., 2004). The warming from the present climate leads to a decrease in ice extent by 35–40$\times10^3$ km$^2$ °C$^{-1}$ air temperature change (Leppäranta, 2023). For our study period, these changes could be attributed to global warming and climate change as well. The average air temperature during the study duration has decreased approx. 0.6–0.7 °C (Source: NASA/GISS; https://climate.nasa.gov/vital-signs/global-temperature/), and the average max extent has decreased from approx. $141\times10^3$ km$^2$ (33.6 % of the total Baltic Sea) during the preceding period to $109\times10^3$ km$^2$ (25.9 %) during the recent period.

## 7 Conclusion

The study focused on analyzing the Baltic Sea's ice characteristics, utilizing satellite and model datasets. The main findings from the study are given below.

⬦ The sea ice statistics (sea ice extent, season length, and mean thickness) from the Copernicus Marine Baltic Sea Physics Reanalysis product (BALTICSEA_MULTIYEAR_PHY_003_011) have been compared with satellite and SAR & Ice charts based statistics. A sea ice fraction threshold of 0.20 for the model dataset provides the most optimal match against



the satellite's 0.15 threshold, minimizing bias and root mean square error for the temporal evolution of sea ice extent over the study period. For the ice thickness, compared to SAR & ice charts, the model overestimates its value roughly by a linear factor of 1.814 in the study.

◇ During the study period, the model dataset consistently predicts an earlier onset of sea ice but aligns better with satellite data regarding the end of the ice season. Notable disparities exist in the Bothnian Bay and Gulf of Finland sub–basins, where the model forecasts longer ice seasons and more sea ice days than the satellite data.

◇ Recent trends showcase decreasing sea ice fraction and thickness in almost every part of the Baltic Sea. Specifically in the Bothnian Bay, the melting phase exhibits a more rapid decrease in ice thickness per percentage decrease in ice fraction, compared to the freezing phase. In the Gulf of Finland sub–basin the mean ice thickness has become less than half of its value during the melting phase during the recent half (2007/08–2020/21) of the study.

◇ The annual mean ice thickness has been decreasing at the highest rate (in absolute terms) in the Bothnian Bay sub–basin with some northern areas reaching up to approx. 0.4 cm/year, but relative trends of ice thickness thinning have been nearly uniform (approx. 5 %) across all of the Baltic Sea sub–basins.

*Code availability.* The codes are available by request via e-mail at shakti.singh@taltech.ee.

*Author contributions.* SS performed the statistical analysis with input and supervision from IM and RU. RU and IM secured the funding for the study. SS wrote the first draft, with input from IM and RU. All authors contributed to discussions in writing this paper.

*Competing interests.* The authors declare that they have no conflict of interest.

*Acknowledgements.* This work was supported by the NOCOS_DT project (NOrdic CryOSphere Digital Twin, VNF23024), funded by the Nordic Council of Ministers and by AdapEST project (Implementation of national climate change adaptation activities in Estonia, VEU23019), funded by EU's LIFE programme. The authors are grateful to the Copernicus Marine Service for providing the datasets.



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
