# Peer review of "Sea ice in the Baltic Sea during 1993/94–2020/21 ice seasons from satellite observations and model reanalysis"

_EGUsphere, 2024_

## Referee Comment (RC1)

**Sea ice in the Baltic Sea during 1993/94–2020/21 ice seasons from satellite observations and model reanalysis**

Shakti Singh, Ilja Maljutenko, and Rivo Uiboupin

**General comments**

In this paper the authors investigated the sea ice characteristics of the Baltic Sea using Copernicus satellite and model reanalysis data products. Primary focus was on assessing the performance of the model reanalysis product in estimating ice season evolution compared to the satellite dataset. The specific objectives as stated by the authors were (a) finding the sea ice fraction threshold to bias correct the model reanalysis dataset for Sea Ice extent (SIE) and comparing ice thickness statistics between the model and SAR & ice charts–based datasets; (b) comparing Baltic Sea reanalysis product's sea ice season evolution characteristics with satellite dataset; (c) analyzing the characteristics and changes of ice extent and thickness in during 1993/94–2020/21; (d) providing trend analysis of the sea ice season parameters and sea ice thickness. It was found that a sea ice fraction threshold of 0.20 for the model dataset provides the most optimal match against the satellite data 0.15 threshold, minimizing bias and root mean square error for the temporal evolution of sea ice extent over the study period. The model estimates an earlier start to the ice season, but it generally matches satellite data regarding the season's end. It was found that the model tends to overestimate ice thickness compared to ice chart-based data. Across the Baltic Sea, declining trends for the sea ice were observed. The sea ice characteristics during the recent period (2007-2021) compared to preceding one (1993-2007) show decreased sea ice fraction and thickness. The decrease in the sea ice thickness was over 50 % in some areas during the melting phase (March to May). In general, there was uniform pattern towards shorter ice seasons, reduced sea ice extent (SIE) and reduced mean ice thickness.

While methodology and results in the paper are properly presented and discussed, I have some questions and remarks on the used datasets and some other major comments below. Satellite data sea ice fraction was taken Copernicus SST product. According to product documentation this sea ice fraction comes from the ice chart in which it based on manual analysis. Thus, sea ice fraction does not come from any automatic algorithm, which must mentioned in the paper.

Sea ice thickness seems to be taken from Copernicus Marine Baltic Sea–Sea Ice Concentration and Thickness Charts product dataset with product id SEAICE_BAL_SEAICE_L4_NRT_ OBSERVATIONS_011_004. This dataset is the manual ice chart, and ice thickness is level ice thickness, and that is why you likely observed model overestimating ice thickness; more off-shore than near the coasts where there is level landfast ice. You refer ice thickness as the SAR & ice chart based product, but ice thickness in the ice chart is based on in-situ data from coastal stations and icebreakers. Or is there an error in the text and you in reality use Baltic Sea - SAR Sea Ice Thickness and Drift, Multisensor Sea Ice Concentration product which has sea ice thickness dataset where ice thickness from the ice chart is spatially enhanced using SAR imagery?

Section 4 should be under Section 5 'Results', and Section 5 could start with short introduction of the section contents.

The sea ice fraction threshold for calculating sea ice extent from the model data was tuned against the satellite data SIE. How do you know here that the satellite data SIE is more accurate than the model SIE? Possible that satellite data needs bias correction not model data?

In Section 4.2 explain how corrected ice thickness was calculated, diving by 1.814?

How results of the trend analysis would change if you use different time intervals, e.g. five or ten years?

Paper by Ronkainen et al. "Interannual sea ice thickness variability in the Bay of Bothnia", The Cryosphere, 12, 3459–3476, 2018, may have results relevant to your study.

**Specific comments**

line 7: "recent period (2007–2021)"

    Definition of this recent period here is vague without defining what is the preceding period.

Figure 1 is not introduced and discussed in Section 1. Same color is used for Bothnian Sea and Danish Straits.

l. 28: BACC II Author Team, 2015

    This reference is missing from reference list.

Discussion on Baltic Sea ice in Introduction could include some figures on typical thickness of level ice and thickness of ridges.

l. 42: "Remote sensing techniques have evolved to enhance the efficacy of ice information services"

    Include here some newer references, e.g. papers by Karvonen et al.

l. 55: ice state over previous model products (QUID_REAN).

    Explain QUID_REAN, or give reference.

l. 71: spell out SMHI and FMI when first time used.

l. 80: "The system is forced by ECMWF ERA5 meteorology (ref)."

    give reference

l. 123: "the dataset (the lm function from R)"

    explain this lm function from R

l. 131: "The quality information document (QUID) …"

    Give reference for the QUID document, it is not Karvonen et al. 2007.

Figure 3 is introduced in the text after Figure 4.

Figure 8: very difficult to see black contour lines.

Table 3: maybe SIT data with 0.1 cm accuracy

---

## Referee Comment (RC2)

**General Comments**

The study provides the assessment of the latest Copernicus Marine Service Baltic Sea Physics reanalysis product (BALTICSEA_MULTIYEAR_PHY_003_011) for 1993/94 to 2020/21. They adopt the satellite data and SAR & ice charts as the validation and continue finding that a significant decline in sea ice fraction and thickness, particularly during the melting phase, was observed, with the Bothnian Bay and Gulf of Finland. The study also emphasizes the recent period (2007/08–2020/21) exhibits a shorter ice season and reduced maximum sea ice extent compared to the preceding period (1993/94–2006/07).

I appreciate the criteria for assessment and the clear objectives listed in the Introduction. However, I have several concerns regarding the usage of validation data, the methodology protocol, and some unclear explanations. Therefore, I recommend that the paper undergo major revisions before it can be considered for publication.

**Here are my major comments:**
1. I do have major concern in the period split: why you choose 2007 as the threshold for the date division, please provide some explanation.
2. Another concern is the data usage in SST_BAL_SST_L4_REP_OBSERVATIONS_010_016 (satellite product) and SEAICE_BAL_SEAICE_L4_NRT_OBSERVATIONS_011_004 (SAR & ice charts-based product). How do you consider the uncertainty in the satellite and SAR & ice charts-based product considering you use the satellite product to determine/correct the sea ice fraction threshold in the reanalysis data, it is important to know the accuracy or uncertainty of the satellite product. And when I look at the Table 1, I am also wondering how is the RMSE and Bias look like during 0.15 and 0.25? What about other thresholds, such as 0.18 or 0.23? And since you've showed two criteria for threshold selection, how do you coordinate them together, such as in RMSE, 0.20 reanalysis threshold has the lowest value while in Bias, 0.25 seems to have the lowest value. And I am quite lost in Line 151, when you mentioned, "TH_SIF of 0.15 for the model dataset, provides more accurate estimates of maximum SIE", can you provide more clearly and statistically evidences in why 0.15 the accurate estimate of maximum SIE is. In Section 4.2, when you are trying to correct the reanalysis sea ice thickness based on three years SAR images and ice chart product, I am not sure if it is statistically robust. Given that samplings for grid is large, but when you consider the annual changes, 3 years is quite short, and not long enough to support your ice thickness correction statements. When I look at the Figure 5, it is quite obviously that Model SIT has the saturation stage in high value compared with the SAR images and ice chart. Then (1) how to explain this condition; (2) instead of the linear relationship, how about using the exponential lines to picture the fitting? And I don't understand how to apply the correction coefficient in Line 168, did you overall divide the values by the 1.81?
3. My next concern is the motivation behind the assessment of the Baltic Sea ice product, which is missing in the Discussion section. For example, what are the limitations of using the current data? What insights can be provided to modelers for improving models? Which updates have improved the product performance compared to previous versions? The current discussion lacks depth and does not provide the audience and the community with sufficient information beyond the assessment results.

**Detailed comments:**

1. Line 65, Dataset part: please provide detailed information on the temporal resolution and time span of the three products. Additionally, explain how you coordinate these products with different resolutions and specify the interpolation methods used.
2. Line 80, please fill in the reference.
3. Figure 2, I suggest moving either panels (a) and (b) or (c) and (d) to the appendix, as they seem to replicate information.
4. Figure 10: specify the units in panels (a) and (d). I'm quite interested in the fitting process in panels (c) and (f). When focusing on the density plot, consider showing how the linear fitting looks when focusing on high-intensity values or averaging bin values, and then performing the linear fitting.
5. Line 231: could you provide an interpretation of why the Gulf of Finland sub-basin exhibits the most significant reduction during the melting season compared to the freezing season?
6. Figure 11 and 12, could you overlay the trend with the 95% significance level? For example, use stipples to indicate the 95% confidence level or plot only the trends that are above the 95% confidence level.
7. Figure 12(b): verify the values around 55°N, 21°E, and explain why this area shows the largest reduction in ice thickness.
8. Line 261, wrong reference format.

---

## Author Comment (AC1)

**Sea ice in the Baltic Sea during 1993/94–2020/21 ice seasons from satellite observations and model reanalysis**

**Anonymous Referee #1,**

https://editor.copernicus.org/index.php?_mdl=msover_md&_jrl=778&_lcm=oc108lcm109w&_acm=get_comm_sup_file&_ms=120859&c=269726&salt=5798643141918097389

Thank you very much for taking the time to thoroughly review the manuscript and for providing your valuable feedback. We have carefully considered your comments and are pleased to provide our responses to your questions and remarks below.

**General comments:**

Rev1.1)

*While methodology and results in the paper are properly presented and discussed, I have some questions and remarks on the used datasets and some other major comments below. Satellite data sea ice fraction was taken Copernicus SST product. According to product documentation this sea ice fraction comes from the ice chart in which it based on manual analysis. Thus, sea ice fraction does not come from any automatic algorithm, which must mentioned in the paper.*

Response: Description "The sea ice fraction data in the product is taken from the sea ice charts from the national ice services at FMI and SMHI, which is based on manual interpretation by the operator" will be added in the manuscript.

Rev1.2)

*Sea ice thickness seems to be taken from Copernicus Marine Baltic Sea–Sea Ice Concentration and Thickness Charts product dataset with product id SEAICE_BAL_SEAICE_L4_ NRT_OBSERVATIONS_011_004. This dataset is the manual ice chart, and ice thickness is level ice thickness, and that is why you likely observed model overestimating ice thickness; more off-shore than near the coasts where there is level landfast ice. You refer ice thickness as the SAR & ice chart based product, but ice thickness in the ice chart is based on in-situ data from coastal stations and icebreakers. Or is there an error in the text and you in reality use Baltic Sea - SAR Sea Ice Thickness and Drift, Multisensor Sea Ice Concentration product which has sea ice thickness dataset where ice thickness from the ice chart is spatially enhanced using SAR imagery?*

Response: There is no error regarding the sea ice thickness product. The product used in the study for sea ice thickness is indeed from Copernicus Marine Baltic Sea–Sea Ice Concentration and Thickness Charts product dataset with product id SEAICE_BAL_SEAICE_L4_ NRT_OBSERVATIONS_011_004. The product has daily ice thickness data from the ice chart which is adjusted using SAR image, mentioned in the product user manual[1] . It is validated against icebreaker ice thickness data in the QUID file of the product[2], and shows similar statistics to the Baltic Sea - SAR Sea Ice Thickness and Drift, Multisensor Sea Ice Concentration[3].

References (for above text):

1. For Baltic Sea – Sea Ice Observations SEAICE_BAL_SEAICE_L4_NRT_ OBSERVATIONS_011_004/011, Issue: 2.14, section III.1.1 Ice charts of the Finnish Ice Service
2. QUID for SEA ICE TAC Products 011_004, 011_011, 011_019, Ref. CMEMS-SEAICE-QUID-004_011_019, issue 1.1
3. Product id: SEAICE_BAL_SEAICE_L4_NRT_OBSERVATIONS_011_011, https://doi.org/10.48670/moi-00133

Rev1.3)

*Section 4 should be under Section 5 'Results', and Section 5 could start with short introduction of the section contents.*

Response: Suggested changes will be made in the manuscript.

Rev1.4)

*The sea ice fraction threshold for calculating sea ice extent from the model data was tuned against the satellite data SIE. How do you know here that the satellite data SIE is more accurate than the model SIE? Possible that satellite data needs bias correction not model data?*

Response: The sea ice data in the satellite product is based on Copernicus SI-TAC products (which includes Ice charts and SAR based sea ice data) and high resolution SMHI data [RD.5] (Product user manual: CMEMS-SST-PUM-010-016-040, https://doi.org/10.48670/moi-00156). Although SIE based on this data can certainly have uncertainties, these datasets have consistently been used in studies to validate other datasets (Karvonen 2021; Leppäranta 2023; Pärn et al. 2021; Mäkynen et al. 2020) and are available for a long term period.

Since there is not yet a better automatic tool for interpretation of sea ice in high resolution imagery, and due to the lack of field observations, ice charts are considered being the data sets that best describes the true state of the sea ice, and thus other sea ice data sets use these as reference. (CMEMS-SI-QUID-011-001to007-009to014, QUALITY INFORMATION DOCUMENT

For SI TAC Sea Ice products 011-001, -002, -004, -006, -007, -009, -010, -011, -012, -013, -014, issue 2.10).

Required changes will be made in the manuscript to make it more clear.

Rev1.5)

*In Section 4.2 explain how corrected ice thickness was calculated, diving by 1.814?*

Response: Yes, that is correct.

It will be mentioned in the manuscript.

Rev1.6)

*How results of the trend analysis would change if you use different time intervals, e.g. five or ten Years?*

Response: In our study, the trends patterns for different time intervals are still somewhat similar with varying magnitude but due to smaller sample size, the trends are not significant at 95 percent confidence interval. Example figure for trend of last 10 years mean sea ice thickness (cm) attached below.

[Figure]

**Specific Comments:**

Rev1.7)

*line 7: "recent period (2007–2021)"*
*Definition of this recent period here is vague without defining what is the preceding period. Figure 1 is not introduced and discussed in Section 1. Same color is used for Bothnian Sea and Danish Straits.*

Response: Suggested changes will be made in the manuscript.
The relevant line in the abstract will be changed to "The sea ice characteristics during the recent period (2007/08–2020/21) show decreased sea ice fraction and thickness compared to the preceding period (1993/94–2006/2007) of the study."
Figure 1 will be introduced and discussed. Color used for the Danish Straits sub-basin will be changed.

Rev1.8)

*l. 28: BACC II Author Team, 2015*
*This reference is missing from reference list.*

Response: Reference "Rasool, S. I., Menenti, M., and Bolle, H.-J.: Second assessment of climate change for the Baltic Sea basin, Springer, 2015" will be added in the manuscript.

Rev1.9)
*l. 42: "Remote sensing techniques have evolved to enhance the efficacy of ice information services"*
*Include here some newer references, e.g. papers by Karvonen et al.*

Response: Newer references "Karvonen 2017; Karvonen 2021" will be added in the manuscript.

Rev1.10)
*l. 55: ice state over previous model products (QUID_REAN).*
*Explain QUID_REAN, or give reference.*

Response: Reference "(Panteleit et al. 2019, CMEMS-BAL-QUID-003-011)" will be added in the manuscript.

Rev1.11)
*l. 71: spell out SMHI and FMI when first time used.*

Response: Suggested changes will be made in the manuscript.

Rev1.12)
*l. 80: "The system is forced by ECMWF ERA5 meteorology (ref)."*
*give reference*

Response: Reference "(Hersbach et al. 2020)" will be added in the manuscript.

Rev1.13)
*l. 123: "the dataset (the lm function from R)"*
*explain this lm function from R*

Response: Description "(the lm function from R, lm is used to fit linear models and can be used to carry out regression)" will be added in the manuscript.

Rev1.14)
*l. 131: "The quality information document (QUID) …"*
*Give reference for the QUID document, it is not Karvonen et al. 2007.*

Response: Reference "(Panteleit et al. 2019, CMEMS-BAL-QUID-003-011)" will be added in the manuscript.

Rev1.15)

*Figure 3 is introduced in the text after Figure 4.*

Response: Required changes will be made in the manuscript.

Rev1.16)

*Figure 8: very difficult to see black contour lines.*

Response: Required changes will be made to improve visibility for the figure.

Rev1.17)

*Table 3: maybe SIT data with 0.1 cm accuracy*

Response: Suggested changes will be made in the manuscript.

---

## Author Comment (AC2)

**Sea ice in the Baltic Sea during 1993/94–2020/21 ice seasons from satellite observations and model reanalysis**

**Anonymous Referee #2**

https://editor.copernicus.org/index.php?_mdl=msover_md&_jrl=778&_lcm=oc108lcm109w&_acm=get_comm_sup_file&_ms=120859&c=270362&salt=1013438105556683178

We greatly appreciate your thorough review of our manuscript and the valuable feedback you provided. We have carefully addressed your comments and are happy to present our responses, along with additional analysis to support our study.

**General comments:**

Rev2.1)
*I do have major concern in the period split: why you choose 2007 as the threshold for the date division, please provide some explanation.*

Response: According to Pärn et al. 2022, A new regime of sea ice began in 2008 and in all basins of the Baltic Sea, a rapid warming during spring could be detected. Hence the study period is divided into two halves. It is mentioned in paper as well "Based on the regime shift detected by Pärn et al. (2022) we split our study period into two parts: 1993/94–2006/07 (preceding period) and 2007/08–2020/21 (recent period)" at line 206.

Rev2.2)
*Another concern is the data usage in SST_BAL_SST_L4_REP_OBSERVATIONS_010_016 (satellite product) and SEAICE_BAL_SEAICE_L4_NRT_OBSERVATIONS_011_004 (SAR & ice charts-based product). How do you consider the uncertainty in the satellite and SAR & ice charts-based product considering you use the satellite product to determine/correct the sea ice fraction threshold in the reanalysis data, it is important to know the accuracy or uncertainty of the satellite product.*

Response: The sea ice data in the satellite product is based on Copernicus SI-TAC products (which includes Ice charts and SAR based data) and high resolution SMHI data [RD.5] (CMEMS-SST-PUM-010-016-040, Product user manual: https://doi.org/10.48670/moi-00156). Although Ice charts based sea ice data can certainly have uncertainties, these datasets have consistently been used in studies to validate other products (Karvonen 2021; Leppäranta 2023; Pärn et al. 2021; Mäkynen et al. 2020) and are available for a long term period.
The product SEAICE_BAL_SEAICE_L4_NRT_OBSERVATIONS_011_004 is validated against icebreaker ice thickness data in the QUID file of the product (QUID for SEA ICE TAC Products 011_004, 011_011, 011_019, Ref. CMEMS-SEAICE-QUID-004_011_019, issue 1.1).
Since there is not yet a better automatic tool for interpretation of sea ice in high resolution imagery, and due to the lack of field observations, ice charts are considered being the data sets that best describes the true state of the sea ice, and thus other sea ice data sets use these as reference.

(CMEMS-SI-QUID-011-001to007-009to014, QUALITY INFORMATION DOCUMENT For SI TAC Sea Ice products 011-001, -002, -004, -006, -007, -009, -010, -011, -012, -013, -014, issue 2.10)
Required changes will be made in the manuscript to make it more clear.

Rev2.3)
*And when I look at the Table 1, I am also wondering how is the RMSE and Bias look like during 0.15 and 0.25? What about other thresholds, such as 0.18 or 0.23? And since you've showed two criteria for threshold selection, how do you coordinate them together, such as in RMSE, 0.20 reanalysis threshold has the lowest value while in Bias, 0.25 seems to have the lowest Value.*

Response: Bias is used as the primary criteria to select optimal threshold, RMSE and CC are calculated to confirm the results. Results from other thresholds are given below. The selection is made out of traditionally used ice fraction thresholds which are generally factors of 0.05 like 0.15, 0.20 etc., especially considering 0.20 gives reasonably low bias.

| TH_SIF (Satellite) | TH_SIF (Model) | Bias (km$^2$) | RMSE (km$^2$) | CC |
|---|---|---|---|---|
| **0.15** | **0.15** | **1783** | **4887** | **0.994** |
| 0.15 | 0.16 | 1490 | 4648 | 0.995 |
| 0.15 | 0.17 | 1208 | 4439 | 0.995 |
| 0.15 | 0.18 | 935 | 4260 | 0.995 |
| 0.15 | 0.19 | 671 | 4110 | 0.995 |
| **0.15** | **0.20** | **416** | **3991** | **0.995** |
| 0.15 | 0.21 | 168 | 3900 | 0.996 |
| 0.15 | 0.22 | -73 | 3837 | 0.996 |
| 0.15 | 0.23 | -309 | 3803 | 0.996 |
| 0.15 | 0.24 | -539 | 3795 | 0.996 |
| **0.15** | **0.25** | **-764** | **3810** | **0.996** |

Due to formatting, the negative sign for bias values corresponding to 0.25 model TH_SIF was missing in the table which will be added. Required changes will be made to reduce ambiguity for the selection criteria.

Rev2.4)
*And I am quite lost in Line 151, when you mentioned, "TH_SIF of 0.15 for the model dataset, provides more accurate estimates of maximum SIE", can you provide more clearly and statistically evidences in why 0.15 the accurate estimate of maximum SIE is.*

Response: It can be seen from figure 4 for max SIE. In terms of Statistics, Bias and RMSE are calculated between max SIE time series (similar to figure 3) for these two threshold combinations and shown below in the table. From the table, it is apparent that TH_SIF of 0.15 for the model dataset provides more accurate estimates of maximum SIE compared to TH_SIF of 0.20.

| TH_SIF (Satellite) | TH_SIF (Model) | Bias (km$^2$) | RMSE (km$^2$) |
|---|---|---|---|
| 0.15 | 0.15 | -1165 | 14682 |
| 0.15 | 0.20 | -5552 | 16143 |

Rev2.5)
*In Section 4.2, when you are trying to correct the reanalysis sea ice thickness based on three years SAR images and ice chart product, I am not sure if it is statistically robust. Given that samplings for grid is large, but when you consider the annual changes, 3 years is quite short, and not long enough to support your ice thickness correction statements.*

Response: Beside it being the common period available between the two datasets, the model reanalysis sea ice thickness data shows considerable variations within these 3 years and probability distribution plot shows similar thickness distribution for these 3 years compared to the complete 27 years reanalysis thickness data.

[Figure]

Required changes will be made to mention it, in the manuscript.

Rev2.6)
*When I look at the Figure 5, it is quite obviously that Model SIT has the saturation stage in high value compared with the SAR images and ice chart. Then (1) how to explain this condition; (2) instead of the linear relationship, how about using the exponential lines to picture the fitting?*

Response: (1) The SAR & ice chart based SIT does seem to get saturated at lower values compared to Model SIT, this could be related to how the data processing and filtering for the SAR & ice charts data is done. Required changes will be made to mention this drawback in the manuscript. The thickness distribution for the common data from SAR & ice charts based and model based data is given below.

[Figure]

(2) Using non linear fitting slightly improves the fit (Example for nonlinear fit shown below), but the difference of parameters such as R squared value is really small compared to linear fit. So to simplify the analysis, linear fitting is chosen (the below equation can be used to get a slightly better fit).

[Figure]

$$y = 0.35 + 2.38x - 0.02x^2$$
R-squared: 0.909

Rev2.7)
*And I don't understand how to apply the correction coefficient in Line 168, did you overall divide the values by the 1.81?*

Response: Yes, that is correct.
It will be mentioned in the manuscript.

Rev2.8)
*My next concern is the motivation behind the assessment of the Baltic Sea ice product, which is missing in the Discussion section. For example, what are the limitations of using the current data? What insights can be provided to modelers for improving models? Which updates have improved the product performance compared to previous versions? The current discussion lacks depth and does not provide the audience and the community with sufficient information beyond the assessment Results.*

Response: Suggested changes will be made in the discussion section of the manuscript.

**Detailed comments:**

Rev2.9)
*Line 65, Dataset part: please provide detailed information on the temporal resolution and time span of the three products. Additionally, explain how you coordinate these products with different resolutions and specify the interpolation methods used.*

Response: The products are interpolated to a common grid system, using bilinear interpolation. In case of satellite and model data, the common grid resolution used is 0.0277 x 0.02, while for Ice charts & SAR and model data, the common grid used is 0.0277 ° × 0.0166 ° (2 × 2 km).
Required description about temporal resolution and extent will be added in the manuscript.

Rev2.10)
*Line 80, please fill in the reference.*

Response: Reference "(Hersbach et al. 2020)" will be added in the manuscript.

Rev2.11)
*Figure 2, I suggest moving either panels (a) and (b) or (c) and (d) to the appendix, as they seem to replicate information.*

Response: Suggested changes will be made in the manuscript.

Rev2.12)
*Figure 10: specify the units in panels (a) and (d). I'm quite interested in the fitting process in panels (c) and (f). When focusing on the density plot, consider showing how the linear fitting looks when focusing on high-intensity values or averaging bin values, and then performing the linear fitting.*

Response: Figure 10: panel (a) and (d) are ice fraction differences hence unitless, for panel (b) and (e) units are specified in figure caption. Figure for Linear fitting when focusing on high intensity values (ice thickness difference more than 15 cm) is shown below (For MAM density plot). The higher

thickness difference values here correspond to mostly the Bothnian Bay basin where thickness changes have been the largest during spring (Fig. 10e).

[Figure]

Rev2.13)

*Line 231: could you provide an interpretation of why the Gulf of Finland sub-basin exhibits the most significant reduction during the melting season compared to the freezing season?*

Response: Although our study has not focused on the underlying reasons for these changes, it is evident that recent research has detected rapid warming during the spring season (Pärn et al. 2022). Given that the Gulf of Finland is a shallow basin, this warming could have contributed to the significant reduction in ice fraction observed during the melting phase (spring season). Required text will be added in the manuscript.

Rev2.14)

*Figure 11 and 12, could you overlay the trend with the 95% significance level? For example, use stipples to indicate the 95% confidence level or plot only the trends that are above the 95% confidence level.*

Response: Changes will be made to overlay the trend with the 95% significance level.

Rev2.15)

*Figure 12(b): verify the values around 55°N, 21°E, and explain why this area shows the largest reduction in ice thickness.*

Response: Around 55°N, 21°E (Curonian lagoon), the absolute trend values for ice thickness are really low for the area, but in terms of percentage of reduction the larger because the area has very thin ice and is separated by Curonian Spit from the Baltic Sea. But we are not concerned about this region in the study hence it is not mentioned in the manuscript.

Rev2.16)
*Line 261, wrong reference format.*

Response: Reference format for "Åström et al. 2024" will be corrected in the manuscript.

---

## Author Response (AR1)

**Sea ice in the Baltic Sea during 1993/94–2020/21 ice seasons from satellite observations and model reanalysis**

**Anonymous Referee #1,**

https://editor.copernicus.org/index.php? mdl=msover md& jrl=778& lcm=oc108lcm109w& acm=get comm sup file& ms=120859&c=269726&salt=5798643141918097389

Thank you very much for taking the time to thoroughly review the manuscript and for providing your valuable feedback. We have carefully considered your comments and are pleased to provide our responses to your questions and remarks below.

**General comments:**

**Rev1.1)**

While methodology and results in the paper are properly presented and discussed, I have some questions and remarks on the used datasets and some other major comments below. Satellite data sea ice fraction was taken Copernicus SST product. According to product documentation this sea ice fraction comes from the ice chart in which it based on manual analysis. Thus, sea ice fraction does not come from any automatic algorithm, which must mentioned in the paper.

Response: Description "The sea ice fraction data in the product is taken from the sea ice charts from the national ice services at FMI and SMHI, based on manual interpretation by the operator" has been added in the manuscript at lines 74-76.

**Rev1.2)**

Sea ice thickness seems to be taken from Copernicus Marine Baltic Sea-Sea Ice Concentration and **Thickness** Charts product dataset with product id SEAICE BAL SEAICE L4 NRT\_OBSERVATIONS\_011\_004. This dataset is the manual ice chart, and ice thickness is level ice thickness, and that is why you likely observed model overestimating ice thickness; more off-shore than near the coasts where there is level landfast ice. You refer ice thickness as the SAR & ice chart based product, but ice thickness in the ice chart is based on in-situ data from coastal stations and icebreakers. Or is there an error in the text and you in reality use Baltic Sea - SAR Sea Ice Thickness and Drift, Multisensor Sea Ice Concentration product which has sea ice thickness dataset where ice thickness from the ice chart is spatially enhanced using SAR imagery?

Response: There is no error regarding the sea ice thickness product. The product used in the study for sea ice thickness is indeed from Copernicus Marine Baltic Sea—Sea Ice Concentration and Thickness Charts product dataset with product id SEAICE\_BAL\_SEAICE\_L4\_ NRT\_OBSERVATIONS\_011\_004. The product has daily ice thickness data from the ice chart which is adjusted using SAR image, mentioned in

the product user manual1. It is validated against icebreaker ice thickness data (Berglund et al., 2014) in the QUID file of the product2, and shows similar statistics to the Baltic Sea - SAR Sea Ice Thickness and Drift, Multisensor Sea Ice Concentration3.

**References (for above text):**

- 1. For Baltic Sea Sea Ice Observations SEAICE\_BAL\_SEAICE\_L4\_NRT\_ OBSERVATIONS\_011\_004/011, Issue: 2.14, section III.1.1 Ice charts of the Finnish Ice Service
- 2. QUID for SEA ICE TAC Products 011\_004, 011\_011, 011\_019, Ref. CMEMS-SEAICE-QUID-004\_011\_019, issue 1.1
- 3. Product id: SEAICE\_BAL\_SEAICE\_L4\_NRT\_OBSERVATIONS\_011\_011, https://doi.org/10.48670/moi-00133

Description "The ice parameters in the product are based on SAR image and ice chart produced on a daily basis during the Baltic Sea ice season provided by FMI and SMHI. The product is validated against icebreaker ice thickness data (Berglund et al., 2014) in the Quality Information Document file (CMEMS-SEAICE-QUID-004 011 019, issue 1.1)" has been added at lines 95-97

**Rev1.3)**

Section 4 should be under Section 5 'Results', and Section 5 could start with short introduction of the section contents.

Response: Suggested changes are made with the added introduction "This section is divided into several subsections, including a comparison of the model with satellite sea ice data for ice extent and thickness; an analysis of sea ice season parameters; a statistical examination of sea ice fraction versus sea ice thickness; and a trend analysis of sea ice characteristics" at lines 146-148.

**Rev1.4)**

The sea ice fraction threshold for calculating sea ice extent from the model data was tuned against the satellite data SIE. How do you know here that the satellite data SIE is more accurate than the model SIE? Possible that satellite data needs bias correction not model data?

Response: The sea ice data in the satellite product is based on Copernicus SI-TAC products (which includes Ice charts and SAR based sea ice data) and high resolution SMHI data [RD.5] (Product user manual: CMEMS-SST-PUM-010-016-040, https://doi.org/10.48670/moi-00156). Although SIE based on this data may have uncertainties, these datasets have consistently been used in studies to validate other datasets such as model products (Karvonen 2021; Leppäranta 2023; Pärn et al. 2021; Mäkynen et al. 2020) and are available for a long term period.

The uncertainties have been highlighted in the QUID file (QUID for SEA ICE TAC Products 011\_004, 011 011, 011 019, Ref. CMEMS-SEAICE-QUID-004 011 019, issue 1.1): The daily ice chart Sea ice

concentration (SIC) was compared with the radiometer product of University of Bremen, which showed a difference of  $\pm$  10%. Taking into account the restrictions of the radiometer product (poor/moderate resolution, mixed pixels) (Spreen et al., 2008), the agreement was good. The ice chart-based ice thickness results were compared with ice thickness results measured on Finnish and Swedish icebreakers. The locations of the IB measurements during the season 2022-2023 were mostly over the Gulf of Bothnia where most of the ice was located and only a few measurements were made in the northern parts of the Gulf of Finland. Over 214 observations, Bias and RMSD value were 5 cm and 9.8 cm respectively.

Since there is not yet a better automatic tool for interpretation of sea ice in high resolution imagery, and due to the lack of systematic long term field observations, ice charts are considered being the data sets that best describes the true state of the sea ice, and thus other sea ice data sets use these as reference. (CMEMS-SI-QUID-011-001to007-009to014, QUALITY INFORMATION DOCUMENT

For SI TAC Sea Ice products 011-001, -002, -004, -006, -007, -009, -010, -011, -012, -013, -014, issue 2.10).

Description "Although ice chart-based sea ice data may have some uncertainties, these datasets have been reliably used in numerous studies to validate other products such as model products (Mäkynen et al., 2020; Karvonen, 2021; Pärn et al., 2021; Leppäranta, 2023) and are available over an extended period. Due to the lack of a better automated tool for interpreting high-resolution sea ice imagery and the limited field observations, ice charts are considered the most accurate datasets for representing the true state of sea ice, and thus other sea ice data sets use them as reference (CMEMS-SI-QUID-011-001to007-009to014, issue 2.10)" has been added at lines 99-103.

**Rev1.5)**

In Section 4.2 explain how corrected ice thickness was calculated, diving by 1.814?

Response: Yes, that is correct.

Description "The corrected ice thicknesses (by dividing the model ice thickness values by factor 1.814)" has been added at lines 192-193.

**Rev1.6)**

How results of the trend analysis would change if you use different time intervals, e.g. five or ten Years?

Response: In our study, the trends patterns for different time intervals are still somewhat similar with varying magnitude but due to smaller sample size, the trends are not significant at 95 percent confidence interval. Example figure for trend of last 10 years mean sea ice thickness (cm) attached below.

**Figure R1.6:** Linear trend of ice thickness (in cm/year) for Nov–May mean ice thickness from 2011/12 to 2020/21. Black dots mark the areas with 95 % significance level

**Specific Comments:**

**Rev1.7)**

line 7: "recent period (2007-2021)"

Definition of this recent period here is vague without defining what is the preceding period. Figure 1 is not introduced and discussed in Section 1. Same color is used for Bothnian Sea and Danish Straits.

Response: Relevant description in abstract has been changed to "The sea ice characteristics during the recent period (2007/08–2020/21) show decreased sea ice fraction and thickness compared to the preceding period (1993/94–2006/07) of the study" at lines 6-8.

Figure 1 is introduced with "The geographical boundaries of the sub-basins used in current study are based on the PLC–6 project, obtained from Helsinki Commision (HELCOM, 2018), shown in Fig. 1 " and discussed at lines 104-105. Color used for the Danish Straits sub-basin has been changed in Fig 1.

**Rev1.8)**

I. 28: BACC II Author Team, 2015

This reference is missing from reference list.

Response: Reference "Rasool, S. I., Menenti, M., and Bolle, H.-J.: Second assessment of climate change for the Baltic Sea basin, Springer, 2015" is added in the manuscript at line 499.

**Rev1.9)**

I. 42: "Remote sensing techniques have evolved to enhance the efficacy of ice information services" Include here some newer references, e.g. papers by Karvonen et al.

Response: Newer references "Karvonen 2017; Karvonen 2021" are added in the manuscript at lines 43-44.

**Rev1.10)**

I. 55: ice state over previous model products (QUID\_REAN). Explain QUID\_REAN, or give reference.

Response: Reference "(Panteleit et al. 2019, CMEMS-BAL-QUID-003-011)" is added in the manuscript at line 56.

**Rev1.11)**

I. 71: spell out SMHI and FMI when first time used.

Response: SMHI and FMI are spelled out when used for the first time at lines 73-74.

**Rev1.12)**

I. 80: "The system is forced by ECMWF ERA5 meteorology (ref)." give reference

Response: Reference "(Hersbach et al. 2020)" is added in the manuscript at line 85.

**Rev1.13)**

I. 123: "the dataset (the Im function from R)" explain this Im function from R

Response: Description "(the Im function from R, Im is used to fit linear models and can be used to carry out regression)" is added in the manuscript at lines 137-138.

**Rev1.14)**

I. 131: "The quality information document (QUID) ..."

Give reference for the QUID document, it is not Karvonen et al. 2007.

Response: Reference "(Panteleit et al. 2019, CMEMS-BAL-QUID-003-011)" is added in the manuscript at line 150.

Rev1.15)

Figure 3 is introduced in the text after Figure 4.

Response: Relevant part has been restructured to introduce figure 3 first in the manuscript at lines 175-176.

Rev1.16)

Figure 8: very difficult to see black contour lines.

Response: Instead of contour lines, discrete color shading has been used to improve visibility for the figure 8.

Rev1.17)

Table 3: maybe SIT data with 0.1 cm accuracy

Response: SIT data in table 3 is rounded off to one decimal point.

Verdana fonts have been used in all the figures for better visibility.

**Sea ice in the Baltic Sea during 1993/94–2020/21 ice seasons from satellite observations and model reanalysis**

**Anonymous Referee #2**

https://editor.copernicus.org/index.php? mdl=msover md& jrl=778& lcm=oc108lcm109w& acm= get comm sup file& ms=120859&c=270362&salt=1013438105556683178

We greatly appreciate your thorough review of our manuscript and the valuable feedback you provided. We have carefully addressed your comments and are happy to present our responses, along with additional analysis to support our study.

**General comments:**

**Rev2.1)**

I do have major concern in the period split: why you choose 2007 as the threshold for the date division, please provide some explanation.

Response: According to Pärn et al. 2022, a new regime of sea ice began in 2008 and in all basins of the Baltic Sea, a rapid warming during spring could be detected. Hence the study period is divided into two halves. It is mentioned in paper as well "Based on the regime shift detected by Pärn et al. (2022) we split our study period into two parts: 1993/94–2006/07 (preceding period) and 2007/08–2020/21 (recent period)" at lines 236-237.

**Rev2.2)**

Another concern is the data usage in SST\_BAL\_SST\_L4\_REP\_OBSERVATIONS\_010\_016 (satellite product) and SEAICE\_BAL\_SEAICE\_L4\_NRT\_OBSERVATIONS\_011\_004 (SAR & ice charts-based product). How do you consider the uncertainty in the satellite and SAR & ice charts-based product considering you use the satellite product to determine/correct the sea ice fraction threshold in the reanalysis data, it is important to know the accuracy or uncertainty of the satellite product.

Response: The sea ice data in the satellite product is based on Copernicus SI-TAC products (which includes lice charts and SAR based data) and high resolution SMHI data [RD.5] (CMEMS-SST-PUM-010-016-040, Product user manual: https://doi.org/10.48670/moi-00156). Although Ice charts based sea ice data may have uncertainties, these datasets have consistently been used in studies to validate other products such as model products (Karvonen 2021; Leppäranta 2023; Pärn et al. 2021; Mäkynen et al. 2020) and are available for a long term period.

The uncertainties have been highlighted in the QUID file (QUID for SEA ICE TAC Products 011\_004, 011\_011, 011\_019, Ref. CMEMS-SEAICE-QUID-004\_011\_019, issue 1.1): The daily ice chart Sea ice concentration (SIC) was compared with the radiometer product of University of Bremen, which showed a difference of  $\pm$  10%. Taking into account the restrictions of the radiometer product (poor/moderate resolution, mixed pixels) (Spreen et al., 2008), the agreement was good. The ice

chart-based ice thickness results were compared with ice thickness results measured on Finnish and Swedish icebreakers. The locations of the IB measurements during the season 2022-2023 were mostly over the Gulf of Bothnia where most of the ice was located and only a few measurements were made in the northern parts of the Gulf of Finland. Over 214 observations, Bias and RMSD value were 5 cm and 9.8 cm respectively.

Since there is not yet a better automatic tool for interpretation of sea ice in high resolution imagery, and due to the lack of field observations, ice charts are considered being the data sets that best describes the true state of the sea ice, and thus other sea ice data sets use these as reference.

(CMEMS-SI-QUID-011-001to007-009to014, QUALITY INFORMATION DOCUMENT For SI TAC Sea Ice products 011-001, -002, -004, -006, -007, -009, -010, -011, -012, -013, -014, issue 2.10)

Description "Although ice chart-based sea ice data may have some uncertainties, these datasets have been reliably used in numerous studies to validate other products as model products (Mäkynen et al., 2020; Karvonen, 2021; Pärn et al., 2021; Leppäranta, 2023) and are available over an extended period. Due to the lack of a better automated tool for interpreting high-resolution sea ice imagery and the limited field observations, ice charts are considered the most accurate datasets for representing the true state of sea ice, and thus other sea ice data sets use them as reference (CMEMS-SI-QUID-011-001to007-009to014, issue 2.10)" has been added at lines 99-103.

**Rev2.3)**

And when I look at the Table 1, I am also wondering how is the RMSE and Bias look like during 0.15 and 0.25? What about other thresholds, such as 0.18 or 0.23? And since you've showed two criteria for threshold selection, how do you coordinate them together, such as in RMSE, 0.20 reanalysis threshold has the lowest value while in Bias, 0.25 seems to have the lowest Value.

Response: Bias is used as the primary criteria to select optimal threshold, RMSE and CC are calculated to confirm the results. Results from other thresholds are given below. The selection is made out of traditionally used ice fraction thresholds which are generally factors of 0.05 like 0.15, 0.20 etc., especially considering 0.20 gives reasonably low bias.

**Table R2.3:** Comparison of sea ice extent calculated using different threshold values of SIF for the model dataset is shown by means of statistical methods such as correlation coefficients (CC), root mean square error (RMSE), and mean bias

| TH_SIF (Satellite) | TH_SIF (Model) | Bias (km²) | RMSE (km²) | СС    |
|--------------------|----------------|------------|------------|-------|
| 0.15               | 0.15           | 1783       | 4887       | 0.994 |
| 0.15               | 0.16           | 1490       | 4648       | 0.995 |
| 0.15               | 0.17           | 1208       | 4439       | 0.995 |
| 0.15               | 0.18           | 935        | 4260       | 0.995 |
| 0.15               | 0.19           | 671        | 4110       | 0.995 |

| 0.15 | 0.20 | 416  | 3991 | 0.995 |
|------|------|------|------|-------|
| 0.15 | 0.21 | 168  | 3900 | 0.996 |
| 0.15 | 0.22 | -73  | 3837 | 0.996 |
| 0.15 | 0.23 | -309 | 3803 | 0.996 |
| 0.15 | 0.24 | -539 | 3795 | 0.996 |
| 0.15 | 0.25 | -764 | 3810 | 0.996 |

Due to formatting, the negative sign for bias values corresponding to 0.25 model TH\_SIF was missing in the table which is added. Required changes are made to reduce ambiguity for the selection criteria.

Descriptions "Bias is used as the primary criteria to select optimal threshold, RMSE and CC are calculated to confirm the results" and "The selection is based on traditionally used ice fraction thresholds, typically in increments of 0.05, such as 0.15 and 0.20, especially considering 0.20 gives reasonably low bias" have been added at lines 167-168 and lines 171-173 respectively.

**Rev2.4)**

And I am quite lost in Line 151, when you mentioned, "TH\_SIF of 0.15 for the model dataset, provides more accurate estimates of maximum SIE", can you provide more clearly and statistically evidences in why 0.15 the accurate estimate of maximum SIE is.

Response: It can be seen from figure 4 for max SIE. In terms of Statistics, Bias and RMSE are calculated between max SIE time series (similar to figure 3) for these two threshold combinations and shown below in the table. From the table, it is apparent that TH\_SIF of 0.15 for the model dataset provides more accurate estimates of maximum SIE compared to TH\_SIF of 0.20.

**Table R2.4:** Comparison of max sea ice extent calculated using different threshold values SIF for the model dataset is shown by means of statistical methods such as root mean square error (RMSE) and mean bias

| TH_SIF (Satellite) | TH_SIF (Model) | Bias (km²) | RMSE (km²) |
|--------------------|----------------|------------|------------|
| 0.15               | 0.15           | -1165      | 14682      |
| 0.15               | 0.20           | -5552      | 16143      |

Description "When comparing maximum SIE time series, the bias between the satellite TH\_SIF of 0.15 and the model TH\_SIF of 0.15 is -1165  $\rm km^2$  while the bias between the satellite TH\_SIF of 0.15 and the model TH\_SIF of 0.20 is -5552  $\rm km^2$ " is added at lines 176-178.

**Rev2.5)**

In Section 4.2, when you are trying to correct the reanalysis sea ice thickness based on three years SAR images and ice chart product, I am not sure if it is statistically robust. Given that samplings for grid is large, but when you consider the annual changes, 3 years is quite short, and not long enough to support your ice thickness correction statements.

Response: Beside it being the common period available between the two datasets, the model reanalysis sea ice thickness data shows considerable variations within these 3 years and probability distribution plot shows overlapping thickness distribution for these 3 years compared to the complete 27 years reanalysis thickness data.

**Figure R2.5:** Probability distribution of ice thickness at 5 cm intervals for the recent three years of data (2018/19-2020/21), and the whole period available (1993/94-2020/21) from the model reanalysis product

Description "One of the drawbacks of this analysis is the limitation in data availability, which restricts the comparison of ice thickness to only three winters. It's worth noting that the model reanalysis sea ice thickness data for these three years exhibits considerable variations, with the probability distribution plot (not shown) indicating a similar thickness distribution when compared to the complete 27-year reanalysis dataset." is added in the manuscript at lines 196-199.

**Rev2.6)**

When I look at the Figure 5, it is quite obviously that Model SIT has the saturation stage in high value compared with the SAR images and ice chart. Then (1) how to explain this condition; (2) instead of the linear relationship, how about using the exponential lines to picture the fitting?

Response: (1) The SAR & ice chart based SIT does seem to get saturated at lower values compared to Model SIT, this could be related to how the data processing and filtering for the SAR & ice charts data is done. Although above 40 cm ice thickness (where SAR & ice charts based ice thickness data is getting saturated), the model has only 3.4 % of the data, and if excluded, it does not affect the slope much in figure 5.

The thickness distribution for the common data from SAR & ice charts based and model based data is given below.

**Figure R2.6A:** Probability distribution histograms of ice thickness at 5 cm intervals for the recent three years (2018/19-2020/21), from SAR & ice charts-based dataset (left) and model reanalysis dataset (right)

Description "Additionally, it's important to note that ice charts and SAR-based data tend to saturate at lower thickness values" is added at lines 199-200.

(2) Using non linear fitting slightly improves the fit (Example for nonlinear fit shown below), but the difference of parameters such as R squared value is really small compared to linear fit. So to simplify the analysis, linear fitting is chosen (the below equation can be used to get a slightly better fit).

**Figure R2.6B:** Time averaged sea ice thickness (in cm) 2D histograms: SAR & ice charts vs model based dataset for 2018/19-2020/21 period

Rev2.7)

And I don't understand how to apply the correction coefficient in Line 168, did you overall divide the values by the 1.81?

Response: Yes, that is correct.

Description "The corrected ice thicknesses (by dividing the model ice thickness values by factor 1.814)" has been added at lines 192-193.

**Rev2.8)**

My next concern is the motivation behind the assessment of the Baltic Sea ice product, which is missing in the Discussion section. For example, what are the limitations of using the current data? What insights can be provided to modelers for improving models? Which updates have improved the product performance compared to previous versions? The current discussion lacks depth and does not provide the audience and the community with sufficient information beyond the assessment Results.

Response: Description "The discrepancies in ice models may stem from both atmospheric forcing fields and uncertainties related to the parameterization of ocean-ice-atmosphere processes (Vihma and Haapala, 2009). The majority of state of art models still lack a full atmosphere-wave-ice-ocean coupling system, which inherits various approximations and model corrections through data assimilation. While the correction of the physical model SST using satellite data assimilation helps reduce SST errors in the Nemo-Nordic 2.0 model, the onset of ice periods is still delayed by approximately a week (Fig 8c). This might mean that there are discrepancies in cooling of sea temperature or errors in early ice formation processes. Given that ice conditions in the Baltic Sea differ from polar regions (e.g. no multi-year ice), bulk parameterizations for ice thermodynamics and rheology processes should be revised according to local ice conditions and ice model should be calibrated using reliable data sources (Pemberton et al., 2017). Additionally, with the development of fine-resolution ice models in dynamic ice regions, it may become relevant to implement more advanced rheology parameterizations, such as brittle ice rheology (Bordeau et al., 2024; Olason et al., 2022)." is added in the manuscript at lines 338-347.

**References**

"Brodeau, L., Rampal, P., Ólason, E., and Dansereau, V.: Implementation of a brittle sea ice rheology in an Eulerian, finite-difference, C-grid modeling framework: impact on the simulated deformation of sea ice in the Arctic, Geoscientific Model Development, 17, 6051–6082, 2024."

"Haapala, J. J., Ronkainen, I., Schmelzer, N., and Sztobryn, M.: Recent change—Sea ice, Second assessment of climate change for the Baltic Sea basin, pp. 145–153, 2015."

"Olason, E., Boutin, G., Korosov, A., Rampal, P., Williams, T., Kimmritz, M., Dansereau, V., and Samaké, A.: A new brittle rheology and numerical framework for large-scale sea-ice models, Journal of Advances in Modeling Earth Systems, 14, e2021MS002 685, 2022." are added in the manuscript at lines 379-381, 403-404 and 480-481 respectively.

**Detailed comments:**

**Rev2.9)**

Line 65, Dataset part: please provide detailed information on the temporal resolution and time span of the three products. Additionally, explain how you coordinate these products with different resolutions and specify the interpolation methods used.

Response: The products are interpolated to a common grid system, using bilinear interpolation. In case of satellite and model data, the common grid resolution used is  $0.0277 \times 0.02$ , while for Ice charts & SAR and model data, the common grid used is  $0.0277 \times 0.0166$ ° (2 × 2 km).

Descriptions "The product data is available daily from 1 Jan 1982 to 31 Dec 2023", "It offers a comprehensive reanalysis of physical conditions in the Baltic Sea available daily from 1 Jan 1993 to 31 Dec 2021", "This dataset has 1×1 km spatial resolution and is available daily from 1 Jan 2018 to 5 Jun 2024" and "The products are resampled to a common grid system using bilinear interpolation for comparison" are added at lines 76, 82-83, 94-95 and 141-142 respectively.

**Rev2.10)**

Line 80, please fill in the reference.

Response: Reference "(Hersbach et al. 2020)" is added in the manuscript at line 85.

**Rev2.11)**

Figure 2, I suggest moving either panels (a) and (b) or (c) and (d) to the appendix, as they seem to replicate information.

Response: Panels (a) and (b) from figure 2 are moved to the appendix figure named Figure A1.

**Rev2.12)**

Figure 10: specify the units in panels (a) and (d). I'm quite interested in the fitting process in panels (c) and (f). When focusing on the density plot, consider showing how the linear fitting looks when focusing on high-intensity values or averaging bin values, and then performing the linear fitting.

Response: Figure 10: panel (a) and (d) are ice fraction differences hence unitless, for panel (b) and (e) units are specified in figure caption. Figure for Linear fitting when focusing on high intensity values (ice thickness difference more than 15 cm) is shown below (For MAM density plot). The higher thickness difference values here correspond to mostly the Bothnian Bay basin where thickness changes have been the largest during spring (Fig. 10e).

**Figure R2.12:** 2D histograms for the MAM fraction vs thickness differences (the recent period 2007/08–2020/21 compared to preceding period 1993/94–2006/07), shaded according to their frequency. Linear regression line is plotted, only focusing on thickness changes larger than 15 cm.

Rev2.13)

Line 231: could you provide an interpretation of why the Gulf of Finland sub-basin exhibits the most significant reduction during the melting season compared to the freezing season?

Response: Although our study has not focused on the underlying reasons for these changes, it is evident that recent research has detected rapid warming during the spring season (Pärn et al. 2022). Given that the Gulf of Finland is a shallow basin, this warming could have contributed to the significant reduction in ice fraction observed during the melting phase (spring season).

Description "Although our study has not focused on the underlying reasons for these changes, it is evident that recent research has detected rapid warming during the spring season (Pärn et al. 2022). Given that the GoF is a shallow basin, this warming could have contributed to the significant reduction in sea ice observed during the melting phase (spring season)" is added in the manuscript at lines 263-265.

Rev2.14)

Figure 11 and 12, could you overlay the trend with the 95% significance level? For example, use stipples to indicate the 95% confidence level or plot only the trends that are above the 95% confidence level.

Response: 95% significance level in the figure 11 and 12 is overlaid using stipples.

Rev2.15)

Figure 12(b): verify the values around 55°N, 21°E, and explain why this area shows the largest reduction in ice thickness.

Response: Around 55°N, 21°E (Curonian lagoon), the absolute trend values for ice thickness are really low for the area, but in terms of percentage of reduction the larger because the area has very thin ice and is separated by Curonian Spit from the Baltic Sea. But we are not focusing on this region in the study hence it is not mentioned in the manuscript.

Rev2.16)

*Line 261, wrong reference format.*

Response: Reference format for "Aström et al. 2024" is corrected in the manuscript at line 295.

Verdana fonts have been used in all the figures for better visibility.

---

## Referee Report (RR1)

**Sea ice in the Baltic Sea during 1993/94–2020/21 ice seasons from satellite observations and model reanalysis**

Shakti Singh, Ilja Maljutenko, and Rivo Uiboupin

**General comments**

The authors have given proper answers to my comments and questions, and have improved the paper accordingly. I don't have any new comments on the paper.

However, I still think that it is still unclear what SAR and ice chart product was used. The SEAICE\_BAL\_SEAICE\_L4\_NRT\_ OBSERVATIONS\_011\_004 product has manual ice chart which shows level ice thickness. The product SEAICE\_BAL\_SEAICE\_L4\_NRT\_OBSERVATIONS\_011\_011 has ice thickness chart based on SAR and ice chart. From PUM: "The SAR data is used to update the ice information in the IC. The ice regions in the IC are updated according to a SAR segmentation and new ice thickness values are assigned to each SAR segment based on the SAR backscattering and the ice IC thickness range at that location." So this thickness chart shows ice chart's level ice thickness in finer spatial resolution, and that is why you likely observed model overestimating ice thickness; more off-shore than near the coasts where there is level landfast ice.

---

## Referee Report (RR2)

**Sea ice in the Baltic Sea during 1993/94–2020/21 ice seasons from satellite observations and model reanalysis**

bv

Shakti Singh, Ilja Maljutenko, and Rivo Uiboupin

It is very difficult to play with the manuscript when two reviewers have already published the reviews. I will keep decisions of the previous reviewers and will mark it as a major revision, however, I do not like this paper. And I also agree with most of the previous comments. There is one that needs to be changed – the reader is not interested in the software used – so the sentence with the explanation of the function 'R' should be removed.

Firstly, it sounds strange or it is a big issue that the reanalysis of sea ice included in the Copernicus database looks like it is wrong. I have no idea what should be fixed, but based on figure 6, the modeled sea ice thickness is excessively overestimated, which suggests problems in the circulation model or wrong parameterizations in the ice model – sea ice is created at the boundary between the ocean and air or between the ocean and sea ice. However, it should be described in the Copernicus database – somebody paid for those data and it has been accepted. To me, the data in the Copernicus database should be treated as a reference, and such dataset should not be accepted.

In my point of view the SAR and ice charts data set is the best.

The paper presents a simple analysis of the three datasets in the Copernicus database. There is nothing special about the paper except the numbers that could be used by other researchers.

I have only small comments:

The freezing and melting seasons depend on time but also on location, which means that the time depends on location, but in this work, it is divided only in time (DJF and MAM). I think it is a wrong approach and should be fixed.

The SD in the paper differs, and I feel there is a problem with the circulation model which is also visible in figure 6.

---

## Author Response (AR2)

**Sea ice in the Baltic Sea during 1993/94–2020/21 ice seasons from satellite observations and model reanalysis**

Thank you so much for the feedback.

**Rev 1.1:**

The SEAICE\_BAL\_SEAICE\_L4\_NRT\_ OBSERVATIONS\_011\_004 product has manual ice chart which shows level ice thickness. The product SEAICE\_BAL\_SEAICE\_L4\_NRT\_OBSERVATIONS\_011\_011 has ice thickness chart based on SAR and ice chart. From PUM: "The SAR data is used to update the ice information in the IC. The ice regions in the IC are updated according to a SAR segmentation and new ice thickness values are assigned to each SAR segment based on the SAR backscattering and the ice IC thickness range at that location." So this thickness chart shows ice chart's level ice thickness in finer spatial resolution, and that is why you likely observed model overestimating ice thickness; more off-shore than near the coasts where there is level landfast ice.

Response: After getting confirmation from the Copernicus support team, the product SEAICE\_BAL\_SEAICE\_L4\_NRT\_OBSERVATIONS\_011\_004 indeed has level ice thickness. Thank you so much for your feedback on this point.

Required changes are made, figure 7 is removed and model reanalysis ice thickness data is compared with the SAR & Ice charts based level ice thickness with relevant reference paper (i.e. Ronkainen et al., 2018). Hence the model reanalysis ice thickness data is used in the analysis without any correction. So in all the figures (after figure 7) and tables which uses ice thickness data (i.e. Fig.9, Fig. 11, Table 3, Table 4), the model ice thickness data is used without any correction.

**Sea ice in the Baltic Sea during 1993/94–2020/21 ice seasons from satellite observations and model reanalysis**

We appreciate your thorough review and feedback.

**Rev 2.1:**

the reader is not interested in the software used – so the sentence with the explanation of the function 'R' should be removed.

Response: The sentence "(the Im function from R, Im is used to fit linear models and can be used to carry out regression)" is removed.

**Rev 2.2:**

Firstly, it sounds strange or it is a big issue that the reanalysis of sea ice included in the Copernicus database looks like it is wrong. I have no idea what should be fixed, but based on figure 6, the modeled sea ice thickness is excessively overestimated, which suggests problems in the circulation model or wrong parameterizations in the ice model – sea ice is created at the boundary between the ocean and air or between the ocean and sea ice. However, it should be described in the Copernicus database – somebody paid for those data and it has been accepted. To me, the data in the Copernicus database should be treated as a reference, and such dataset should not be accepted. The SD in the paper differs, and I feel there is a problem with the circulation model which is also visible in figure 6.

Response: The issue is resolved, as after clarification from the Copernicus support team, the product SEAICE\_BAL\_SEAICE\_L4\_NRT\_OBSERVATIONS\_011\_004 has level ice thickness as opposed to total ice thickness. Hence values are much lower than the model dataset.Relevant reference (Ronkainen et al., 2018) is added for the comparison. So the model reanalysis ice thickness data is used in the analysis without any correction.

The SAR & Ice charts data is only consistently available for a short period as compared to the model reanalysis data, which is consistent and available for the whole study duration. Also ice thickness in the model when compared against the level ice thickness from SAR & ice charts, are in the reasonable range indicated by earlier study by Ronkainen et al., 2018).

The Copernicus model reanalysis data is compared with the global ocean and sea-ice reanalysis (ORAS5: Ocean Reanalysis System 5) dataset, which is the monthly mean sea-ice reanalysis data prepared by the European Centre for Medium-Range Weather Forecasts (ECMWF) OCEAN5 ocean analysis-reanalysis system. The comparison between the two products shows similar ice thickness values (see Fig. rev2.2 below). The mean bias of Baltic sea physics reanalysis dataset is -3.7 cm, against the ORAS5 dataset. Thus, the Copernicus products have similar quality as the ECMWF product.

**Fig.** rev2.2: Sea ice thickness: Global ocean and sea-ice reanalysis (ORAS5: Ocean Reanalysis System 5) vs Baltic sea physics reanalysis, from 1993 to 2014. Dotted red line (1:1) is added for reference.

**Rev 2.3:**

The freezing and melting seasons depend on time but also on location, which means that the time depends on location, but in this work, it is divided only in time (DJF and MAM). I think it is a wrong approach and should be fixed.

Response: The ice season was divided into DJF and MAM periods to study changes during these two seasons separately, as previous studies (such as Pärn et al., 2022) have inferred rapid warming during spring (MAM) season. We agree the terminology of freezing and melting season was not correct and hence it is not used anymore, and they are referred to as winter and spring season instead throughout the whole paper.

Freezing and melting periods (which are defined as the periods before and after the max ice thickness at each grid respectively) statistics are provided below, incase of interest (Fig. rev2.3.1 and Fig. rev2.3.2).

**Fig.** rev2.3.1: Spatial map for date of transition from freezing to melting phase averaged over all ice seasons, the freezing and melting phases are the ice periods before and after the max ice thickness respectively

**Fig.** rev2.3.2: The changes of the SIF and SIT during the recent period 2007/08–2020/21 compared to preceding period 1993/94–2006/07, Panels (a) and (d) show the sea ice fraction (SIF) differences; (b) and (e) the sea ice thickness differences (SIT, in m); while (c) and (f) shows the 2D histograms plotted for the fraction vs thickness differences, shaded according to their frequency. Upper panels are for the freezing period and lower panels are for the melting period.

**Rev2.4**

The paper presents a simple analysis of the three datasets in the Copernicus database. There is nothing special about the paper except the numbers that could be used by other researchers.

Response: Thank you for your feedback. While it is true that the paper provides numerical data and statistics, we believe the study offers new valuable insights for the ice climatology for the whole Baltic Sea and its sub-basins, and has important implications for the scientific community, policymakers, and industries operating in the Baltic Sea region.

The operations in this region are greatly influenced by the presence of sea ice. Accurate modeling and forecasting of sea ice seasons are essential for ensuring safe navigation, particularly for maritime transport, fishing fleets, and commercial activities such as the construction of offshore wind farms. Our study provides detailed spatio-temporal insights into ice conditions, enabling better planning and risk management for the stakeholders of these sectors. The influence of sea ice on the Baltic Sea ecosystem is significant, as ice duration impacts the timing of spring blooms, phytoplankton composition, nutrient cycles, and overall ecosystem dynamics. These changes are directly tied to the fisheries industry, and our study helps to quantify the decrease in sea ice, offering insights into its ecological implications.

The research also contributes to a broader understanding of environmental and climatic changes in the region. By providing detailed analysis on seasonal ice coverage patterns (e.g ice thickness and coverage relationship on Fig 9), the study highlights fluctuations in ice extent and long-term shifts in environmental conditions, offering valuable context for understanding regional climate trends. In contrast to earlier studies, which focused primarily on coastal observations, our research extends the analysis to offshore areas for the updated period 1993/94 to 2020/21. This expanded scope reveals a notable reduction in sea ice, particularly in offshore areas, reflecting the impact of recent global warming trends.

Moreover, the detailed analysis and validation of sea ice conditions provided in this study across various spatio-temporal scales, is relevant for the development and usage of the open datasets provided by Copernicus or Destination Earth platforms. Integrating this improved sea ice information into sea ice Digital Twins enables more accurate "what-if" scenario analyses, enhancing the planning and management of offshore activities.

Finally, this study addresses critical gaps in current knowledge about sea ice conditions in the Baltic Sea which are valuable insights for further sea ice modelling and validation analysis. Recent research on this topic has been limited, and discrepancies in existing ice models highlight the need for updated and thorough analyses like ours.

---

## Author Response (AR4)

**Sea ice in the Baltic Sea during 1993/94–2020/21 ice seasons from satellite observations and model reanalysis**

We appreciate the thorough reading and constructive feedback.

**Rev 4:**

More details into either physical mechanism or technique processes will benefit the improvement of the current MS. Besides, the content in the discussion section is strange because they are more suitable for the introduction section than here.

Response: We have revised the manuscript accordingly. In the discussion section, we have added further details on the physical mechanisms underlying sea ice decline, supported by relevant literature and additional analyses (e.g., the relationship with atmospheric patterns such as the NAO and AO).

To address model improvement, we have highlighted relevant approaches from the literature that are particularly applicable to the Baltic Sea and connect them to the key issues identified in our analysis (e.g., biases in onset timing and spring thinning rates).

Some of the discussion part, which was more suitable for the introduction,, has been moved to the introduction section.